# Investigating the Influence of Biochar Amendment on the Physicochemical Properties of Podzolic Soil

**Ratnajit Saha** [1,*], **Lakshman Galagedara** [1,2] , **Raymond Thomas** [2] , **Muhammad Nadeem** [2] **and Kelly Hawboldt** [3]

1   Environmental Science Program, Faculty of Science, Memorial University of Newfoundland, St. John's, NL A1B 3X7, Canada; lgalagedara@grenfell.mun.ca
2   School of Science and the Environment, Memorial University of Newfoundland, Corner Brook, NL A2H 5G4, Canada; rthomas@grenfell.mun.ca (R.T.); mnadeem@grenfell.mun.ca (M.N.)
3   Department of Process Engineering, Memorial University of Newfoundland, St. John's, NL A1C 5S7, Canada; khawboldt@mun.ca
*   Correspondence: saharatnajit@gmail.com; Tel.: +1-709-691-7550

**Abstract:** Research into biochar, as an amendment to soil, has increased over the last decade. However, there is still much to understand regarding the effects of biochar type and rates on the physicochemical properties of different soil types. This study aimed to investigate the effects of biochar application on the physicochemical properties of podzolic soils. Soil samples were collected from the research site in Pasadena, Newfoundland, Canada. Experimental treatments consisted of three types of soils (topsoil, E-horizon soil and mixed soil (topsoil 2: E-horizon soil 1)), two biochar types (granular and powder) and four biochar application rates (0%, 0.5%, 1% and 2% on a weight basis). Ten physicochemical parameters (bulk density (BD), porosity, field capacity (FC), plant available water (PAW), water repellency (WR), electrical conductivity (EC), pH, cation exchange capacity (CEC), total carbon (TC), and nitrogen (N)) were investigated through a total of 72 experimental units. Biochar morphological structure and pore size distribution were examined using a scanning electron microscope, whereas specific surface area was assessed by the Brunauer–Emmett–Teller method. The result indicated that the E-horizon soil was highly acidic compared to control (topsoil) and mixed soils. A significant difference was observed between the control and 2% biochar amendment in all three soil mixtures tested in this experiment. Biochar amendments significantly reduced the soil BD (E-horizon: 1.40–1.25 > mixed soil: 1.34–1.21 > topsoil: 1.31–1.18 g cm$^{-3}$), increased the CEC (mixed soil: 2.83–3.61 > topsoil: 2.61–2.70 > E-horizon: 1.40–1.25 cmol kg$^{-1}$) and total C (topsoil: 2.40–2.41 textgreater mixed soil: 1.74–1.75 > E-horizon: 0.43–0.44%). Water drop penetration tests showed increased WR with increasing biochar doses from 0 to 2% (topsoil: 2.33–4.00 > mixed soil: 2.33–3.33 > E-horizon: 4.00–4.67 s), and all the biochar–soil combinations were classified as slightly-repellent. We found significant effects of biochar application on soil water retention. Porosity increased by 2.8%, FC by 10%, and PAW by 12.9% when the soil was treated with powdered biochar. Additionally, we examined the temporal effect of biochar (0 to 2% doses) on pH and EC and observed an increase in pH (4.3–5.5) and EC (0.0–0.20 dS/m) every day from day 1–day 7. Collectively the study findings suggest 2% powder biochar application rate is the best combination to improve the physicochemical properties of the tested mixed podzolic soil. Granular and powdered biochar was found to be hydrophobic and hydrophilic, respectively. These findings could be helpful to better understand the use of biochar for improving the physicochemical properties of podzolic soils when used for agricultural practices in boreal ecosystems.

**Keywords:** granular biochar; powder biochar; biochar rates; topsoil; E-horizon soil; mixed soil

## 1. Introduction

For more than a decade, research into the diverse applications of biochar has been increasing [1]. This increase is primarily driven by potential applications for biochar in agricultural practices [2–6]. Biochar refers to the solid carbonaceous product [7] produced naturally and commercially [8] via pyrolysis and gasification [9] from organic biomass. Biochar production process temperature range between 300 and 1000 °C under minimum or no oxygen conditions [10,11]. A variety of feed stocks are considered as source material for biochar production including residues produced in agricultural, forestry, and municipal operations [12,13].

Biochar has been well studied as a soil amendment [6,13] and some of these study findings suggest biochar can improve soil physicochemical properties. For example, increasing values of cation exchange capacity (CEC) was reported [14], soil water interaction, and retention (including degraded soils) [15], saturated hydraulic conductivity [16], soil aggregation and stability [17] following soil amendment with biochar. Furthermore, biochar has also been reported to influence soil health by enhancing microbial abundance and diversity in the rhizosphere, as well as modifying the favorable microbial environments by adjusting nutrients, increasing soil pH [18–20] and modifying plant growth performance through improved fertility, and direct supply of carbon (C) rich substrates [21,22]. Biochar application or amendment is also well recognized as a useful potential climate change mitigation strategy [23,24]. Specifically, this mitigation strategy includes decreased erosion potential [13], remediation of contaminated water or soil [25–27], and improved agricultural production and ecosystem sustainability [10,28].

Biochar application may have a positive or negative impact on soil texture and soil hydrological properties such as field capacity (FC) and plant available water (PAW) [8,29] depending on whether the biochar applied is hydrophilic or hydrophobic. If biochar has hydrophobic characteristics, then soil nutrient distribution and retention capacity could be negatively affected, whereas hydrophilic biochar could increase interactions with soil solution [30]. Biochar amendment can influence soil hydraulic properties such as porosity and water retention in three ways: (1) increased total porosity i.e., contributes to pore distribution; (2) modification of compaction between soil and biochar as well as surrounding aggregation; and (3) increased aggregate stability or reduced soil erosion potential by improving pore space persistence [10,31]. Biochar helps to mitigate extreme hydrological conditions (such as very low moisture content (MC)), thus acting as a soil amendment for sustainable farming practices that aim to conserve soil and water [32,33]. Biochar can also positively or negatively impact the soil hydrological properties depending on the intensity of its application [34]. Therefore, characterization of biochar before its application is important as it influences the MC, water holding capacity (WHC), PAW (larger pores retain water weakly under gravity and smaller pores do not have enough space to hold water in a plant-accessible form), ability to absorb water, nutrients, and agricultural chemicals [10,15,35–38]. Pore size distribution is very important to study if biochar is to be used as a soil amendment [10,16]. Determination of pore size and pore distribution pattern from different angles in biochar is essential to understand the effects on WHC and nutrient absorption capacity. Biochar characteristics like surface area and porosity depend on the temperature of pyrolysis and the raw materials used for biochar production [4,39]. Pore sizes can be used to further characterize biochar where pores less than 2 nm in size are referred to as micropores, 2–50 nm as mesopores, and larger than 50 nm as macro-pores [27,40]. Biochar properties such as elemental composition, pH, redox potential (Eh), CEC, and surface functional groups control the interaction between soil and biochar properties [20,41–43] and can modulate soil performance. There is very little information in the current scientific literature on how these modulations affect podzolic soil performance and crop production in the boreal ecosystem.

In the boreal ecosystem, a cool climate with a short crop growing season is considered a critical limitation to agricultural productivity [44–46]. Major factors that result in poor productivity are low soil pH and fertility, and uneven rainfall distribution. Potential toxicity from soluble forms of Al, Mn, and Fe, are also factors that can negatively impact the physical properties of podzolic soils [47]. Sandy soil is prone to high nutrient leaching due to continuous spring, summer and fall

rainfall, and large spring thaw. Innovation in production systems and a better understanding of crop adaptation strategies are therefore critical for sustaining agricultural productivity in boreal regions. Improving soil quality is essential to help ensure food and agricultural quality in a world with a rising population in the boreal regions [48]. Modern scientific literature is replete with examples of biochar's ability to improve soil physicochemical properties and agricultural conditions. However, only a small number of studies have been conducted on the biochar amendment to improve the soil performance including physicochemical properties of podzolic soil in boreal climates or ecosystems [23,49–51]. This is important to note considering the boreal ecosystem is facing an increase in agricultural production and covers approximately 11% of the terrestrial land surface on Earth [48]. In 2019, Wanniarachchi et al. [52] reported that biochar application influences some properties of podzolic soil (e.g., water repellency—WR). However, biochar stability in agricultural practices and the mechanism of alteration of physicochemical properties in acidic soils has not been investigated [30].

Because of the lack of studies on biochar application in acidic soils, we applied biochar on podzolic soil to investigate further. In the agricultural field basically two types of biochar are used: granular and powder biochar. Our overarching hypotheses were "application of biochar to podzolic soil will improve soil hydraulic properties" and "granular biochar is less efficient than powder biochar in improving availability of PAW". The objective of the study was to investigate the influence of biochar amendment on the physicochemical properties of podzolic soil. Overall, the goal of the study was to evaluate the effects of different types of biochar on important soil physicochemical properties that can influence crop performance when cultivated on podzolic soils in boreal ecosystem. We conducted experiments focusing on several research questions such as: What are the effects of different biochar types and application rates on the physicochemical properties of topsoil, E-horizon when separated, and when mixed? Are there any temporal effects of the biochar amendment on the physicochemical properties of podzolic soils (pH, electrical conductivity (EC), MC, and WR)? What are the impacts of granular and powder biochar on the podzol soil water retention curve (SWRC)? How do biochar characteristics (pH, EC, CEC, surface area, pore size, and distribution pattern) influence the physicochemical properties of podzolic soil? We expect our study findings significantly improve our understanding of how biochar application used as a soil amendment and improve soil health, as well as agricultural productivity of podzols, in the boreal ecosystem.

## 2. Materials and Methods

### 2.1. Sampling Site, Sample Collection from Field and Sample Preparation

To conduct the study, we collected soil samples from the agricultural experimental station managed by the Department of Fisheries, Forestry and Agriculture of the Government of Newfoundland and Labrador in Pasadena (49°5′38.63″ N and 57°32′9.32″ W), Newfoundland, Canada, on June 19, 2019. Topsoil and E-horizon soil samples were collected from 0 to 15-cm depth and brought to the Boreal Ecosystems Research Facility (BERF) of Grenfell Campus, Memorial University of Newfoundland. We air-dried the soil samples for seven days at room temperature, then sieved using a 2-mm sieve. During soil sampling from the field, we observed that the average depth of topsoil was around 10 cm, while the below E-horizon soil was around 5 cm thick. Before cultivation, soil is usually ploughed forming horizon consisting a mixed of topsoil and E-horizon. Based on these observations, the mixed soil was prepared combing topsoil and E-horizon soils with a ratio of 2 (topsoil): 1 (E-horizon soil)—according to the average soil layers in the field. Three soil types were evaluated: topsoil, E-horizon soil, and the mixed soil. We used two types of biochar: granular biochar (Market Product—Yellow Pine, Pinus Spp. at 500 °C 30 min, Air Terra Inc., Calgary, AB, Canada), and powdered biochar (Market Product—Maple Hardwood 450 °C, ABRI Tech Inc., Namur, QC, Canada). Soil samples were amended with four biochar rates (weight basis): 0% (control), 0.5% (5 g kg soil), 1% (10 g kg soil), and 2% (20 g kg soil). If we assume the average bulk density (BD) of 1.25 gcm$^{-3}$, and biochar incorporation at a soil depth of 0.1 m, these rates would be equal to 6.25, 12.5 and 25.0 Mg of biochar

per ha. Ten important physicochemical parameters of the soil (BD, porosity, FC, PAW, WR, EC, pH, CEC, total carbon (TC), and nitrogen (N)) were analyzed in this study. A total of 72 experimental units, including all treatments and parameter combinations (3 types of soil, 2 types of biochar with 4 rates, 10 physicochemical properties, and 3 replications), were conducted to assess the effects of biochar on physicochemical properties of podzolic soil (Figure 1).

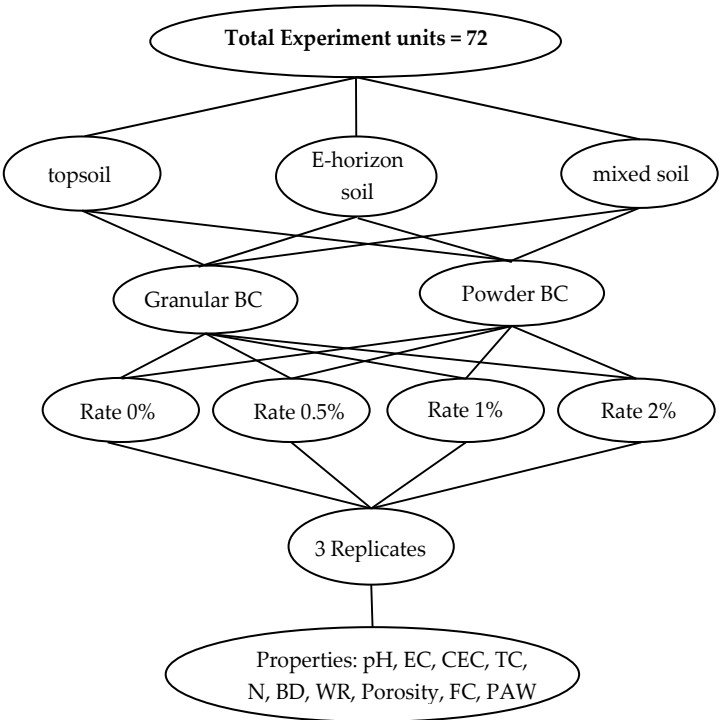

**Figure 1.** Flow chart of investigating the effect of biochar type and different rates on physicochemical properties of podzolic soils. Abbreviations: EC—electrical conductivity; CEC—cation exchange capacity; TC—total carbon; N—nitrogen; BD—bulk density; WR—water repellency; FC—field capacity; PAW—plant available water.

### 2.2. Biochar Characterization

The surface of granular and powder biochar was analyzed using a scanning electron microscope (SEM), to provide detail about physicochemical, elemental, and morphological properties [53,54]. We took the high-resolution images from 1 mm to 2 µm (magnification 100× to 49,999× with the pressure $4.96 \times 10^{-6}$ mbar to $4.67 \times 10^{-6}$ mbar) for both biochar types to identify the diameter and availability of the pore space [13,53]. Biochar specific surface area was analyzed by the Brunauer−Emmett−Teller (BET) $N_2$ ($0.162$ nm$^2$) method and used a NOVA-2000E surface area analyzer [55,56].

### 2.3. Soil Particle Size Analysis

The hydrometer method was used for soil particle size analysis following the procedures of the Bouyoucos method [57]. Topsoil and E-horizon soil samples were analyzed and calculated the particle size in the mixed soil according to the mixing ratio of topsoil (2): E-horizon soil (1). The percentages of clay (<0.002 mm), silt (0.002–0.05mm), and sand (0.05–2 mm) were calculated following Equations (1)–(3).

$$\% \text{ Clay} = \text{Hydrometer reading at 6 h., 52 min.} \times 100/ \text{ wt. of samples} \tag{1}$$

$$\% \text{ Silt} = \text{Hydrometer reading at 40 s.} \times 100/ \text{ wt. of samples} - \% \text{ clay} \tag{2}$$

$$\% \text{ Sand} = 100\% - \% \text{ Silt} - \% \text{ Clay} \tag{3}$$

### 2.4. Bulk Density (BD)

The BD was measured in the samples for each treatment. For this, 86.7 cm$^3$ volume metal containers were filled (tapped three times on the table) with soil and biochar samples (three replications). The samples were oven-dried for 24 h at 105 °C in a forced air-dried oven (SHEL LAB, SHELDON Manufacturing. Inc., Cornelius, NC, USA). Then BD of the samples (dry sample mass/volume of sample) was calculated. The BD of all the treatment combinations were calculated following Equation (4) [52].

$$\text{BD} = \frac{100}{\left[\left(\frac{x}{\rho 1}\right) + \left(\frac{100-x}{\rho 2}\right)\right]} \tag{4}$$

where BD, $\rho 1$, and $\rho 2$ are the bulk densities (gcm$^{-3}$) of the biochar: soil mixtures, biochar only, and soil only, respectively, and $x$ is the biochar rate (%) by weight.

### 2.5. Water Repellency (WR)

We conducted a water drop penetration time (WDPT) test in a 3.85 cm height and 5.81 cm diameter cylindrical plastic container. The bottom of each container was covered by a geotextile cloth to hold the samples. To conduct the WDPT test, about 78.64 ± 0.1 to 92.03 ± 0.1 g of samples were placed in each container. We used the same samples for both WR and BD measurements. A 50-μL volume burette was used in the WDPT test, and one drop of deionized water (50 ± 1 μL) was placed on the top of the sample surface from about a 10-mm height. A stopwatch was used to record the penetration time after placing the water drop. We recorded the time carefully when the drop of water penetrated the surface layer of the sample to ensure accuracy. The WR measurement values were categorized following Leelamanie et al. [58] who suggested the following WR categories: non-repellent (≤1 s), slightly repellent (1–60 s), strongly repellent (60–600 s), severely repellent (600–3600 s) and extremely repellent (≥3600 s). To protect each sample from evaporation during the experiment, we covered each sample container with a lid. To measure the temporal effects, we measured MC at FC and conducted the WDPT test. After that, samples were kept in a forced air-dried oven (SHEL LAB, SHELDON Manufacturing. Inc, Cornelius, OR, USA) at 28 °C, and sample weights were measured from day 1 to day 6 to estimate the MC and conducted the WDPT test.

### 2.6. pH and Electrical Conductivity (EC)

We used a portable pH/EC/TDS/Temperature meter (HANNA—HI9813–6 with CAL Check, ON, Canada) to measure pH and EC in all sample treatments. A 15-g air-dried sample from each biochar-treated soil was diluted in 30 mL of de-ionized water (1:2 ratio) in 50-mL prewashed polypropylene tubes (VWR, Mississauga, ON, Canada). Each vial containing the sample was stirred for one hour at 100 RPM speed. Samples were then left in the vials for 30 min to settle. pH and EC measurements were taken continuously for 7 days, and updated measurements were taken almost at the same time each day [50,59].

### 2.7. Cation Exchange Capacity (CEC)

For all soil amendments, CEC was measured chromatographically by changing sodium acetate with soil or biochar cations [50]. Ion chromatography (Dionex ICS-5000 + DC-5 detector/chromatography module) was used to analyze the amount of sodium concentration in the sample solutions. We observed interference among sodium (Na) and ammonium (NH$^+_4$) peaks, and the values were corrected to get the actual concentration of Na (cmolkg$^{-1}$) using the following Equation (5):

$$\text{Exchangeable amount of Na} = \frac{a \times b \times mcf}{(d) \times (23) \times s} \tag{5}$$

where $a$ = Na concentration (ppm); $mcf$ = the moisture correction factor of oven-dried soil; $s$ = the weight of air-dried sample (g); $b$ = amount of ammonium acetate solution (33 mL); $d$ = the conversion factor (10) from ppm to cmol kg$^{-1}$; 23 = the molecular weight of Na.

## 2.8. Total Carbon (TC) and Nitrogen (N)

TC and N were determined by elemental combustion analysis [41,60]. Three types of soil (topsoil, E-horizon, and mixed soil) and two types of biochar (granular and powder) with three replications were homogenized as follows: the samples were ground well to make the particle size very fine, then around 5 mg samples were taken in the designated capsule and the percentage of TC and N contained in the sample were measured using the CHNS Analyzer (Series II CHNS/O Analyzer 2400, PerkinElmer, Folio Instruments, Inc., CT, USA). Based on the estimated % of TC and N of topsoil, E-horizon soil, granular, and powder biochar, we calculated % of TC and N values for all the combinations using 0, 0.5, 1, and 2% biochar rates.

## 2.9. Soil Water Retention Curve (SWRC)

### 2.9.1. Porosity and Field Capacity (FC)

A pressure plate apparatus system (0700CG23F1 Manifold and 0505V# Compressor, model 1600, Soil Moisture Equipment Corp., Goleta, CA, USA) was used for developing the SWRC. Soil (control) and soil amended with biochar (treatments) weights needed for the sampling ring was calculated based on the predetermined BD, ring volume (21.53 cm$^3$), and the moisture factor of the respective sample. The total weight of each sample (including filter paper, plastic ring, and sample) was taken. Samples were saturated for 3 days using a shallow plastic plate and maintained a water height just below the top of the plastic ring (around 0.2 cm). Once the samples were assumed to reach saturation (when a film of water on the soil surface was observed), sample weights were measured ($Ws$) and samples were then arranged on 50 kPa ceramic plates and placed at the 500 kPa pressure chambers. The initial data point was taken at 10 kPa. Once the water release from the chamber stopped and samples reached a constant weight at the set pressure, the weight of samples was measured. Then samples were placed in the chamber again, and the pressure was increased to 20 kPa. The procedure was repeated subsequently for 30, 40, 50, 70, 100, 300, 400, 500, 600, and 700 kPa pressure levels. The porosity was calculated using Equation (6) at 0 kPa and FC was estimated at 10 kPa using Equation (7) [59].

$$\text{Porosity (p)} = \frac{Ws - Wp}{Vt} \times 100 \tag{6}$$

$$\text{Field Capacity (FC)} = \frac{Wd - Wp}{Vt} \times 100 \tag{7}$$

where $Ws$ is the saturated soil core weight, $Wp$ is the dried soil core weight, $Wd$ is the drained soil core weight and $Vt$ is the core volume.

### 2.9.2. Plant Available Water (PAW)

We converted all gravimetric water contents at each pressure level ($\psi$) to volumetric soil moisture contents (VSMC) ($\theta$) using calculated BD of the samples. Then, we fitted VSMC ($\theta$), and pressure ($\psi$) to the van Genuchten (VG) model [61], Equation (8) [62]. Usually, VG model describes the SWRC of unsaturated soils. We used a VG function to predict how the value would change from 800 to 1500 kPa (Equation (8) using measured data acquired up to 700 kPa. The VSMC at the permanent wilting point (PWP) was predicted using the fitted VG equation, with the pressure being set at 1500 kPa.

$$\theta(\psi) = \frac{(\theta_s - \theta_r)}{\left[1 + \propto (\psi)^n\right]^m} + \theta_r \tag{8}$$

where $\theta(\psi)$ is the VSMC ($cm^3cm^{-3}$) at a given matric potential $\psi(kPa)$; $\theta_s$ is the saturated water content ($cm^3cm^{-3}$) when $\psi$ the 0 kPa; $\theta_r$ is the residual water content ($cm^3cm^{-3}$). At $\psi \geq -1500$ kPa (0.05–0.07) and $\propto$, n, and m shape parameters of the VG equation (Equation (8)). We adopted the Mualem constant (m = 1 − 1/n) to increase model parsimony [28,61,63,64].

We prepared SWRC of measured values vs. predicted values using the VG equation up to PWP at 1500 kPa. Additionally, we created a 1:1 line graph of measured and predicted value to check whether the slope is equal to 1 and the intercept is equal to 0. The root means square error (RMSE) was calculated between estimated and predicted values to check the accuracy of the prediction.

PAW or water storage available for plant use was calculated as the difference between VSMC at field capacity ($\theta_{FC}$) and permanent wilting point ($\theta_{PWP}$) (Equation (9)).

$$\text{Plant Available Water (PAW)} = \theta_{FC} - \theta_{PWP} \tag{9}$$

### 2.10. Statistical Analysis

To assess and quantify the effects of different biochar types and application rates on the physicochemical properties of three soils, we conducted a general analysis of variance (ANOVA). Fisher's least significant difference (LSD) test was used to compare control (only soil without biochar) and treatment groups at alpha = 0.05 to know the significant effects of different variables in different treatments [36]. Descriptive statistical analysis was performed using the software STATA 12.0 version and MS Excel. Graphical visualizations were done through MS excel. Principal Component Analysis (PCA) of data was carried out using XLSTAT (Premium 2017, Version 19.5, Addinsoft, New York, NY, USA).

## 3. Results

### 3.1. Basic Physicochemical Properties of Soil and Biochar

The basic physicochemical properties of these two biochar types are shown in Table 1.

**Table 1.** Physicochemical properties of soil and biochar.

| Physicochemical Parameters | Unit | Topsoil | E-Horizon Soil | Mixed Soil | Granular Biochar | Powder Biochar |
|---|---|---|---|---|---|---|
| pH | | 5.3 ± 0.00 | 4.3 ± 0.00 | 5.6 ± 0.00 | 9 * | 8.9 * |
| EC | $dSm^{-1}$ | 0.15 ± 0.02 | 0.00 ± 0.00 | 0.00 ± 0.00 | 5.2 * | 1.3 * |
| CEC | $cmolkg^{-1}$ | 4.99 ± 0.09 | 2.61 ± 0.27 | 3.61 ± 0.17 | 11.07 ± 0.70 | 5.76 ± 0.31 |
| BD | $gcm^{-3}$ | 1.31 ± 0.10 | 1.40 ± 0.00 | 1.34 ± 0.00 | 0.20 ± 0.00 | 0.35 ± 0.00 |
| WR | s | 1.67 ± 0.58 | 4.00 ± 0.00 | 3.00 ± 0.00 | - | - |
| TC | % | 2.40 ± 0.00 | 0.43 ± 0.00 | 1.74 ± 0.00 | 24.53 ± 1.24 | 58.63 ± 4.60 |
| N | % | 0.05 ± 0.00 | 0.07 ± 0.00 | 0.06 ± 0.00 | 0.04 ± 0.01 | 0.1 ± 0.04 |
| Porosity | % | 52.00 | 39.63 | 48.41 | - | - |
| FC | % | 29.18 | 27.86 | 30.33 | - | - |
| Surface Area | $m^2g^{-1}$ | - | - | - | Almost nonporous | 12.9 |
| Clay | % | 16 | 10 | 14 | - | - |
| Silt | % | 24 | 32 | 27 | - | - |
| Sand | % | 60 | 58 | 59 | - | - |

Note: Top, E-horizon and mixed soil, granular and powder biochar (n = 3) and abbreviations: CEC—cation exchange capacity; EC—electrical conductivity; BD—bulk density; WR—water repellency; TC—total carbon; N—nitrogen; FC—field capacity. * Source: Wanniarachchi et al., 2019b [52].

### 3.2. Characteristics of Granular and Powder Biochar (BET and SEM Image Analysis)

The surface area (SA) of powdered biochar was found to be 12.9 $m^2g^{-1}$, and granular biochar SA was very low (almost non-porous). A high-resolution SEM image (100×–49,999× mag) taken showed morphological and well-defined pore structure and distribution pattern (arrangement of pore space),

helping to differentiate the granular from the powdered biochar (Figures 2 and 3). In granular biochar images, the pore space was found only on one side (Figure 2c), whereas on the other sides, surface walls contained no pore spaces (Figure 2d,e). Some pore space entrances were found to be collapsed with fragments (Figure 2($d_2$,$e_1$)). In the granular biochar samples, pore sizes were not evenly distributed or appeared uniform. Pore spaces were observed on different sides with diameters larger than 10 μm (Figure 2$e_2$). We also observed that pore spaces were blocked (partially) by the fragments (Figure 2f). According to the observation in Figure 2g, pore diameters were found to be <2 μm and 2–5 μm (app.) at different sides of the granular biochar surface, with a non-uniformly distributed number of pores.

The percentage of pore space in powdered biochar was low (Figure 3$c_{1,2}$. Pore spaces were found on only one side, and other sides were occupied by mobile components (like system fragments) in the powder biochar (Figure 3d). On the other side, the surface was not smooth at 5000× magnification with a 30-μm scale (Figure 3$e_{1,2}$). Based on the image scale at different angles, uneven distribution of pore space and size were observed in powdered biochar. The image showed the average pore size was larger than 5 μm (or around 10 μm), with some pores being blocked by the fragments (Figure 3$f_{1,2}$).

Surface morphologies of both granular and powder biochar showed very heterogeneous and structurally complex pore size and shape distribution. In both granular and powder biochar samples, some of the micropores (as visible in the SEM image at different angles) were slightly blocked by the system fragments (as mobile components), damaging micropores. The wall structures of both types of biochar were smooth. Both biochar types surface had a rough surface, even at different angles, perhaps due to the collapse of the pore space becoming filled with system fragments (Figures 2 and 3).

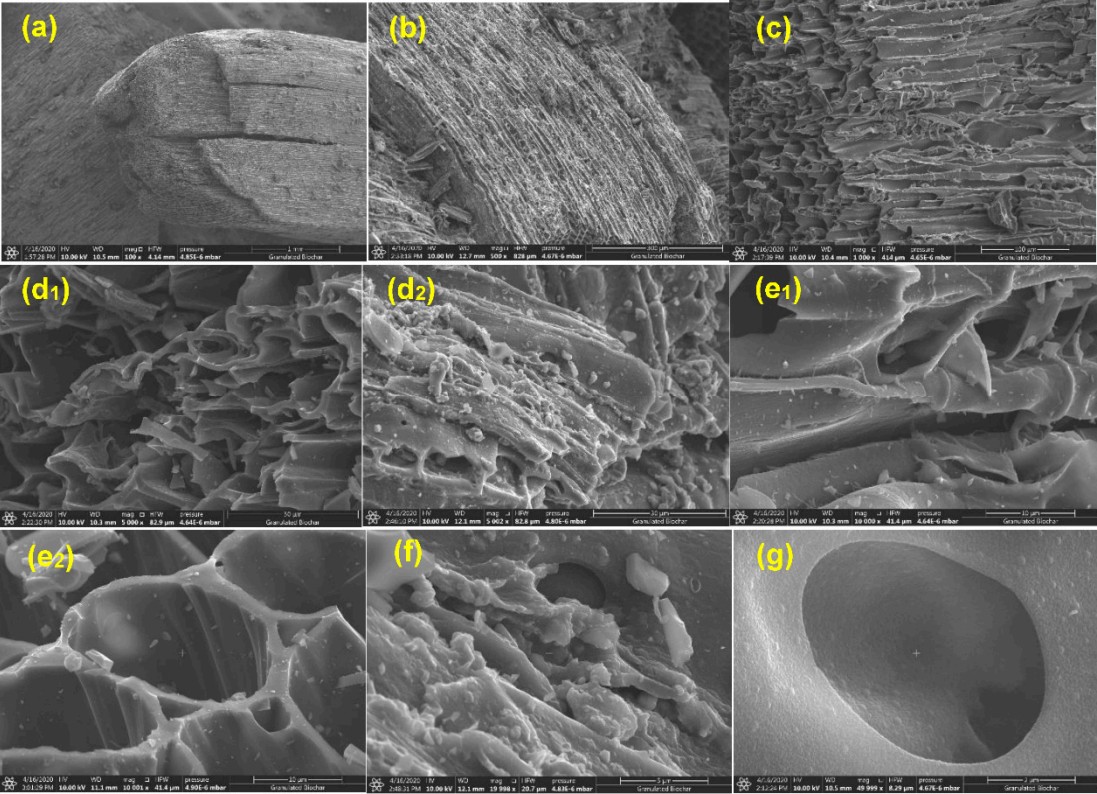

**Figure 2.** Scanning electron microscopy image of granular biochar at different magnifications and scales: (**a**) 100× mag—1-mm scale; (**b**) 500× mag—300-μm scale; (**c**) 1000× mag—100-μm scale; (**d₁**) 5000× mag—30-μm scale; (**d₂**) 5000× mag—30-μm scale; (**e₁**) 10,000× mag—10-μm scale; (**e₂**) 10,001× mag—10-μm scale; (**f**) 19,998× mag—5-μm scale; (**g**) 49,999× mag—2-μm scale.

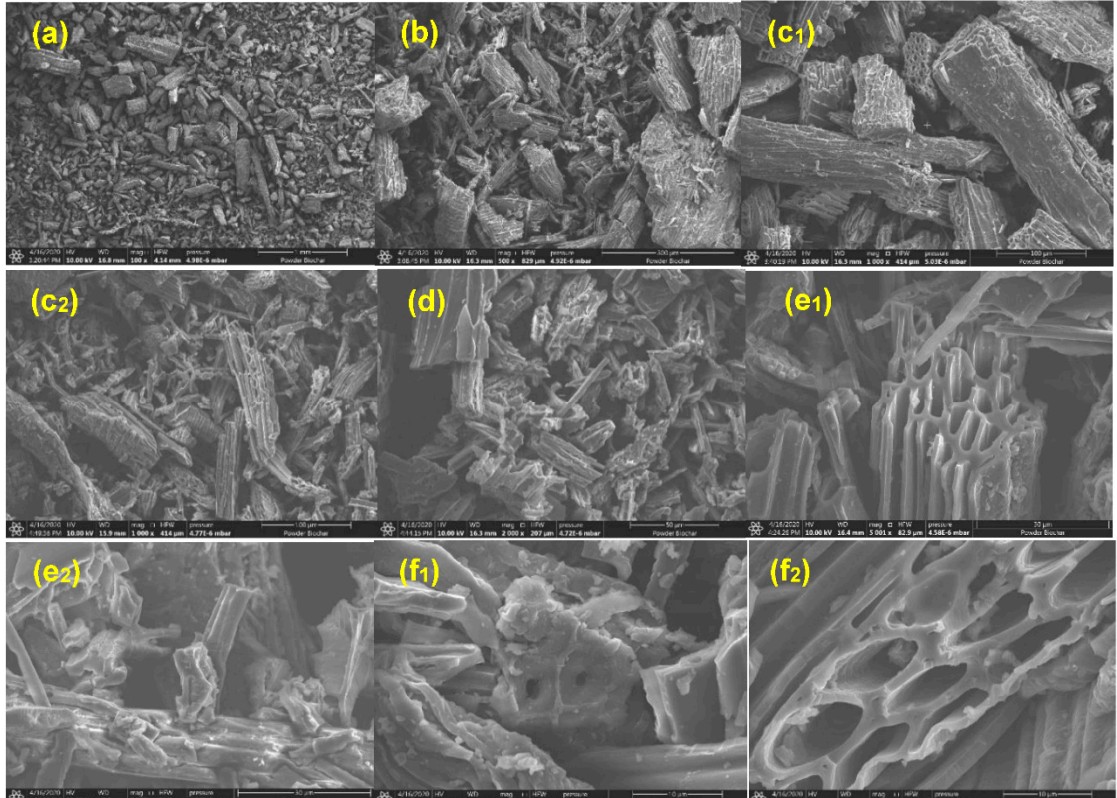

**Figure 3.** Scanning electron microscopy image of powder biochar at different magnification and scale: (**a**) 100× mag—1-mm scale; (**b**) 502× mag—300-μm scale; (**c₁**) 1000× mag—100-μm scale; (**c₂**) 1000× mag—100-μm scale; (**d**) 2000× mag—50-μm scale; (**e₁**) 5001× mag—30-μm scale; (**e₂**) 5000× mag—30-μm scale; (**f₁**) 10,001× mag—10-μm scale; (**f₂**) 10,000× mag—10-μm scale.

### 3.3. Soil Particle Size Distribution

Soil particle size distribution showed a higher percentage of sand-size particles in topsoil (60%) and E-horizon soil (58%) than clay size particles. We found topsoil, E-horizon soil, and mixed soil to be loam, sandy loam, and sandy loam, respectively (Table 1).

### 3.4. Bulk Density (BD)

Biochar types and varying application rates had significant ($p < 0.05$) effects on BD across all three soil combinations. When we applied granular biochar in the topsoil, BD was reduced in the following order: control (1.31 gcm$^{-3}$) > 0.5% (1.28 gcm$^{-3}$) > 1% (1.24 gcm$^{-3}$) > 2% (1.18 gcm$^{-3}$). The same sequence was found (from the calculated BD) when we added powder biochar in the topsoil; control (1.31 gcm$^{-3}$) > 0.5% (1.29 gcm$^{-3}$) > 1% (1.28 gcm$^{-3}$) > 2% (1.24 gcm$^{-3}$). BD trends were highly correlated to applied biochar ratio (Table 2), which was expected. Granular biochar had significant ($p < 0.05$) effects on BD of E-horizon soil (control (1.40 gcm$^{-3}$) > 0.5% (1.36 gcm$^{-3}$) > 1% (1.32 gcm$^{-3}$) > 2% (1.25 gcm$^{-3}$). When applying powder biochar in E-horizon soil, BD also reduced (1.38, 1.36 and 1.32 gcm$^{-3}$ with biochar rate 0.5%, 1% and 2%, respectively) compared to the control (1.40 g cm$^{-3}$). Both granular and powder biochar-amended mixed soil BD were decreased (1.30, 1.27 and 1.21 gcm$^{-3}$ with the granular biochar and 1.32, 1.30, 1.27 gcm$^{-3}$ with the powder biochar rate of 0.5%, 1% and 2%, respectively) when compared to the control (1.34 gcm$^{-3}$).

**Table 2.** Effect of biochar type and application rates on bulk density, water repellency, cation exchange capacity and total carbon, nitrogen on podzolic soil.

| Sl. No | Soil Type | Biochar | | Bulk Density (gcm$^{-3}$) | WR (s) [*p*] | CEC (cmolkg$^{-1}$) [*p*] | Total C (%) | Nitrogen (%) |
|---|---|---|---|---|---|---|---|---|
| 1 | | | 0% | 1.31 ± 0.1 | 1.67 ± 0.58 | 4.99 ± 0.09 | 2.40 ± 0.0 | 0.05 ± 0.0 |
| 2 | | GBC | 0.5% | 1.28 ± 0.0 | 2.33 ± 0.58 [0.39] | 4.92 ± 1.26 [0.20] | 2.40 ± 0.0 | 0.05 ± 0.0 |
| 3 | | | 1% | 1.24 ± 0.0 | 3.00 ± 0.00 | 4.57 ± 0.85 [0.20] | 2.40 ± 0.0 | 0.05 ± 0.0 |
| 4 | | | 2% | 1.18 ± 0.0 | 3.00 ± 0.00 | 5.09 ± 1.76 [0.20] | 2.40 ± 0.0 | 0.05 ± 0.0 |
| 5 | Top soil | | 0% | 1.31 ± 0.1 | 2.33 ± 0.58 | 4.93 ± 0.08 | 2.40 ± 0.0 | 0.05 ± 0.0 |
| 6 | | PBC | 0.5% | 1.29 ± 0.0 | 3.00 ± 0.00 | 5.34 ± 0.60 [0.20] | 2.40 ± 0.0 | 0.05 ± 0.0 |
| 7 | | | 1% | 1.28 ± 0.0 | 2.67 ± 0.58 [0.08] | 5.24 ± 0.10 [0.20] | 2.41 ± 0.0 | 0.05 ± 0.0 |
| 8 | | | 2% | 1.24 ± 0.0 | 4.00 ± 0.00 | 4.99 ± 0.18 [0.20] | 2.41 ± 0.0 | 0.05 ± 0.0 |
| 9 | | | 0% | 1.40 ± 0.0 | 4.00 ± 0.00 | 2.61 ± 0.27 | 0.43 ± 0.0 | 0.07 ± 0.0 |
| 10 | | GBC | 0.5% | 1.36 ± 0.0 | 4.67 ± 0.58 | 2.39 ± 0.34 [0.20] | 0.43 ± 0.0 | 0.07 ± 0.0 |
| 11 | | | 1% | 1.32 ± 0.0 | 4.67 ± 0.58 | 2.35 ± 0.49 [0.20] | 0.43 ± 0.0 | 0.07 ± 0.0 |
| 12 | | | 2% | 1.25 ± 0.0 | 4.67 ± 0.58 | 2.70 ± 0.31 [0.20] | 0.43 ± 0.0 | 0.07 ± 0.0 |
| 13 | E-horizon soil | | 0% | 1.40 ± 0.0 | 4.00 ± 0.00 | 2.28 ± 0.49 | 0.43 ± 0.0 | 0.07 ± 0.0 |
| 14 | | PBC | 0.5% | 1.38 ± 0.0 | 4.33 ± 0.58 | 1.71 ± 0.41 [0.20] | 0.43 ± 0.0 | 0.07 ± 0.0 |
| 15 | | | 1% | 1.36 ± 0.0 | 4.00 ± 0.00 | 1.43 ± 0.06 [0.20] | 0.44 ± 0.0 | 0.07 ± 0.0 |
| 16 | | | 2% | 1.32 ± 0.0 | 4.67 ± 0.58 | 1.53 ± 0.07 [0.22] | 0.44 ± 0.0 | 0.07 ± 0.0 |
| 17 | | | 0% | 1.34 ± 0.0 | 3.00 ± 0.00 | 2.83 ± 1.32 | 1.74 ± 0.0 | 0.06 ± 0.0 |
| 18 | | GBC | 0.5% | 1.30 ± 0.0 | 3.67 ± 0.58 | 3.64 ± 0.45 [0.20] | 1.74 ± 0.0 | 0.06 ± 0.0 |
| 19 | | | 1% | 1.27 ± 0.0 | 3.00 ± 0.00 | 3.49 ± 0.41 [0.20] | 1.74 ± 0.0 | 0.06 ± 0.0 |
| 20 | | | 2% | 1.21 ± 0.0 | 3.00 ± 0.00 | 3.61 ± 0.22 [0.20] | 1.74 ± 0.0 | 0.06 ± 0.0 |
| 21 | Mixed soil | | 0% | 1.34 ± 0.0 | 2.33 ± 0.58 | 3.61 ± 0.17 | 1.74 ± 0.0 | 0.06 ± 0.0 |
| 22 | | PBC | 0.5% | 1.32 ± 0.0 | 2.33 ± 0.58 [0.08] | 3.59 ± 0.15 [0.20] | 1.74 ± 0.0 | 0.06 ± 0.0 |
| 23 | | | 1% | 1.30 ± 0.0 | 3.00 ± 0.00 | 3.38 ± 0.30 [0.20] | 1.75 ± 0.0 | 0.06 ± 0.0 |
| 24 | | | 2% | 1.27 ± 0.0 | 3.33 ± 0.58 [0.39] | 3.35 ± 0.13 [0.20] | 1.75 ± 0.0 | 0.06 ± 0.0 |

Abbreviations: GBC–granular biochar; PBC—powder biochar; WR—water repellency at air dried condition; Total C—total carbon. Note: *p* value—treatment values compared to control. All the treatment combinations of bulk density, total C and nitrogen were calculated values based on experimental values of three types of soil and two types of biochar as mentioned in the method.

*3.5. Water Repellency (WR)*

The WR of granular biochar-amended topsoil slightly increased (2.33, 3.00, and 3.00 s with 0.5%, 1%, and 2% biochar application rates, respectively) compared to the control topsoil (1.67 s). Both granular and powder biochar-amended E-horizon soil increased WR characteristics (4.33, 4.67, 4.67 s when applied with 0.5%, 1%, and 2% biochar, respectively) compared to the control (4.00 s). We found no significant difference for WR when 1% and 2% powder biochar was applied on mixed soil. Overall, when we increased the rate of biochar on the treatment combination, we observed that WR increased with the biochar application rate. The experimental results indicated that the biochar amendment increased small WR (by the order of seconds). All the treatments were classified as a slightly-water repellent, as the WDPT was below 5 s in most of the cases at air-dried conditions.

### 3.6. Cation Exchange Capacity (CEC)

In the experiment, when soil was amended with different types and rates of biochar, topsoil (4.99–5.09 cmolkg$^{-1}$) showed higher CEC values than E-horizon soil (2.61–2.70 cmolkg$^{-1}$) and mixed soil (2.83–3.61 cmolkg$^{-1}$) (Table 2). Both granular and powder biochar had almost no effect on the CEC of E-horizon soil; 2% granular biochar application in the mixed soil showed higher CEC than the control (2.83 cmolkg$^{-1}$). Biochar-amended topsoil slightly increased CEC with high variabilities.

### 3.7. Total Carbon (TC) and Nitrogen (N)

The TC content of granular biochar was 24.53 ± 1.24%, while for powder biochar it was 58.64 ± 0.04% (Table 1). In the experiment, we applied 0.5%, 1%, and 2% of both types of biochar in the podzolic soil. Powder biochar amendment slightly impacted TC content in the treated topsoil. The soil–biochar treatment combinations were found to be 2.40% C with 0.5% biochar, 2.41% C with 1% biochar, and 2.41% C with 2% biochar, with the control topsoil C content being 2.40%. When we applied powder biochar (0–2%) in E-horizon soil, the percentage of C slightly increased from 0.43 to 0.44%. In the case of powder biochar-amended mixed soil, the percentage of C content increased by 1.74% (with 0% biochar) and 1.75% (with 2% biochar).

The sample biochar contained only a low level of nitrogen (N), at 0.04% (granular biochar), and 0.1% (powder biochar) (Table 1). In all types of biochar-treated soils, the N levels increased when the biochar application rate increased.

### 3.8. Soil Water Retention Curve (SWRC)

Figure 4 compares measured and predicted (using VG equation) values in the SWRC. Table 3 outlines the co-efficient of determination (R$^2$) for all the treatment combinations. SWRC showed the changes in the characteristics of three types of soils (top, E-horizon, and mixed soil) after amendment with two biochar types (granular and powder) at different rates (0%, 1%, and 2%) (Figure 4). We observed that the powder biochar amendment increased SWR, compared to the control, and high-water retention capacity compared to granular biochar. In the experiment, important physical parameters such as porosity, FC, and PAW increased with the increasing rates of powder biochar amendment in the podzolic soil.

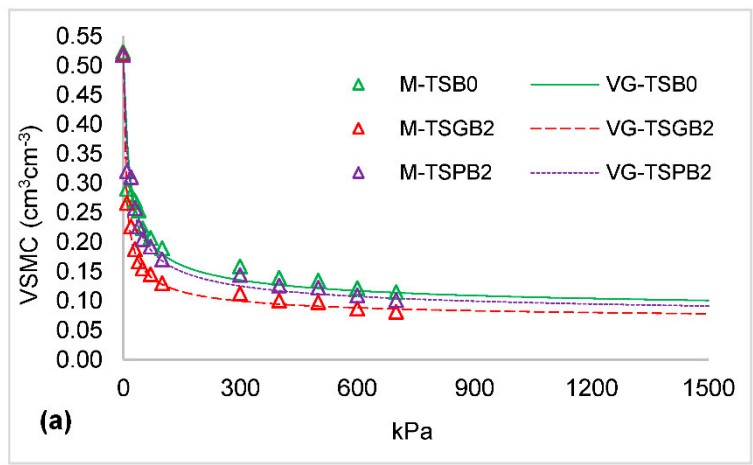

**Figure 4.** *Cont.*

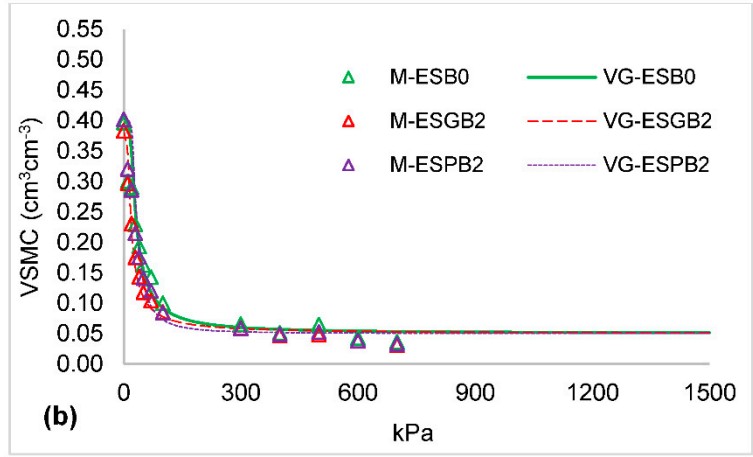

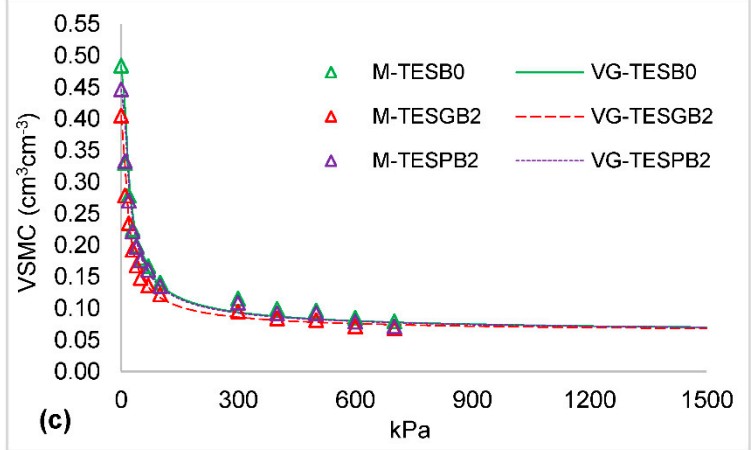

**Figure 4.** Soil water retention curve (SWRC) of measured (M) values vs. predicted values using van Genucthen (VG) equation up to permanent wilting point (PWP) at 1500 kPa for: (**a**) topsoil, (**b**) E-horizon soil and (**c**) mixed soil. Abbreviations: VSMC—volumetric soil moisture content; M-TSB0—measured topsoil with 0% biochar; VG-TSB0—predicted topsoil with 0% biochar; M-TSGB2—measured topsoil with 2% granular biochar; VG-TSGB2—predicted topsoil with 2% granular biochar; M-TSPB2—measured topsoil with 2% powder biochar; VG-TSPB2—predicted topsoil with 2% powder biochar; M-ESB0—measured E-horizon soil with 0% biochar; VG-ESB0—predicted E-horizon soil with 0% biochar; M-ESGB2—measured E-horizon soil with 2% granular biochar; VG-ESGB2—predicted E-horizon soil with 2% granular biochar; M-ESPB2—measured E-horizon soil with 2% powder biochar; VG-ESPB2—predicted E-horizon soil with 2% powder biochar; M-TESB0—measured mixed soil with 0% biochar; VG-TESB0—predicted mixed soil with 0% biochar; M-TESGB2—measured mixed soil with 2% granular biochar; VG-TESGB2—predicted mixed soil with 2% granular biochar; M-TESPB2—measured mixed soil with 2% powder biochar; VG-TSPB2—predicted mixed soil with 2% powder biochar.

**Table 3.** Fitted and measured values of the saturated and residual water content and parameters of the fitted values of the soil water retention model.

| Sl. No | Soil Type | Biochar | | $\theta_s$ (cm$^3$cm$^{-3}$) | | $\theta_r$ (cm$^3$cm$^{-3}$) | $\alpha$ | n | R$^2$ |
| | | Type | Rate | a | b | | | | |
|---|---|---|---|---|---|---|---|---|---|
| 01 | | | 0% | 0.52 | 0.52 | 0.07 | 1.98 | 1.81 | 0.9534 |
| 02 | | GBC | 1% | 0.51 | 0.51 | 0.06 | 3.37 | 1.89 | 0.9889 |
| 03 | Top soil | | 2% | 0.52 | 0.52 | 0.06 | 4.75 | 1.89 | 0.9973 |
| 04 | | | 0% | 0.52 | 0.52 | 0.06 | 1.74 | 1.75 | 0.9549 |
| 05 | | PBC | 1% | 0.53 | 0.53 | 0.06 | 1.95 | 1.68 | 0.9652 |
| 06 | | | 2% | 0.52 | 0.52 | 0.06 | 2.27 | 1.76 | 0.9857 |
| 07 | | | 0% | 0.39 | 0.39 | 0.05 | 0.49 | 5.65 | 0.9421 |
| 08 | | GBC | 1% | 0.39 | 0.39 | 0.05 | 0.45 | 6.95 | 0.9612 |
| 09 | E-horizon | | 2% | 0.38 | 0.38 | 0.05 | 0.90 | 4.85 | 0.9714 |
| 10 | soil | | 0% | 0.40 | 0.40 | 0.05 | 0.49 | 5.65 | 0.9543 |
| 11 | | PBC | 1% | 0.41 | 0.41 | 0.05 | 0.85 | 4.32 | 0.9693 |
| 12 | | | 2% | 0.40 | 0.40 | 0.05 | 0.41 | 8.27 | 0.9514 |
| 13 | | | 0% | 0.48 | 0.48 | 0.06 | 0.95 | 2.87 | 0.9677 |
| 14 | | GBC | 1% | 0.47 | 0.47 | 0.06 | 1.25 | 2.81 | 0.9784 |
| 15 | Mixed soil | | 2% | 0.41 | 0.41 | 0.06 | 1.20 | 2.75 | 0.9851 |
| 16 | | | 0% | 0.44 | 0.44 | 0.06 | 0.90 | 2.85 | 0.9601 |
| 17 | | PBC | 1% | 0.46 | 0.46 | 0.06 | 0.90 | 2.95 | 0.9685 |
| 18 | | | 2% | 0.45 | 0.45 | 0.06 | 0.90 | 2.85 | 0.9826 |

Abbreviations: GBC—granular biochar; PBC—powder biochar; $\theta_s$—the saturated water content (cm$^3$cm$^{-3}$) when $\psi \leq -3$ kPa; $\theta_r$—the residual water content range 0.05–0.07 (cm$^3$cm$^{-3}$) at $\psi \leq -1500$ kPa; $\alpha$, and n—shape parameters of the van Genuchten (1980) model [61]. Van Genuchten parameters were fitted to the data between $\approx$1.8 and 1500 kPa only. The parameters are therefore only valid for this water potential range. a—the measured values of $\theta_s$; b—the fitted values of $\theta_s$.

### 3.8.1. Porosity and Field Capacity (FC)

In the topsoil amended with powder biochar, porosity increased slightly by 1.1% (with 1% biochar rate) and by 1.6% (with 2% biochar rate), compared to the control. In the case of granular biochar application to the topsoil, porosity showed a slightly decreasing trend compared to the control. For example, porosity decreased by 3.1% with a 1% biochar rate. In the powder biochar-treated E-horizon soil, porosity increased by 2.4% (with 1% biochar rate) and by 5.0% (with 2% biochar rate) when compared to the control E-horizon soil. When E-horizon soil was amended with granular biochar, porosity decreased by 4.2% with a 2% granular biochar rate. In the case of mixed soil, porosity increased by 2.8% with a 1% powder biochar rate. When we applied granular biochar in mixed soil, porosity decreased by 2.6% (with 1% biochar rate) and by 16.5% (with 2% biochar rate) when compared to the control mixed soil (Figure 5). The results imply that using powder biochar as a soil amendment would be suitable to increase soil moisture content in all three types of soil.

In the powder biochar-amended topsoil, FC increased by 3.3% with a 1% rate and by 6.7% with a 2% rate compared to control topsoil. In contrast, when we applied granular biochar to the same topsoil, FC decreased by 3.4% (with 1% biochar rate) and by 8.3% (with 2% biochar rate) when compared to the control topsoil. Similarly, when we applied power biochar in E-horizon soil, FC increased by 3.3% (with 1% biochar rate) and by 6.7% (with 2% biochar rate) compared to the control E-horizon soil. However, FC in the granular biochar-amended E-horizon soil decreased slightly by 0.8% (with 1% biochar rate) and by 5.3% (with 2% biochar rate) when compared to the control E-horizon soil. As for the powder biochar-amended mixed soil, FC increased by 4.1% (with 1% biochar rate) and by 10.0% (with 2% biochar rate) when compared to the control mixed soil. In the case of mixed soil amended with granular biochar, FC showed a decreasing trend, decreasing by 6.6% with a 1% biochar rate and by 15.5% with a 2% biochar rate compared to the control mixed soil (Figure 5). These results suggest that the powder biochar amendment in the podzolic soil is found to be suitable in increasing FC. Thus, the WHC in the powder biochar-treated soil was found to be higher than the WHC in the granular

biochar-treated soil. Overall, the experiment results indicated that applied granular biochar possesses hydrophobic characteristics, and powder biochar possesses hydrophilic characteristics.

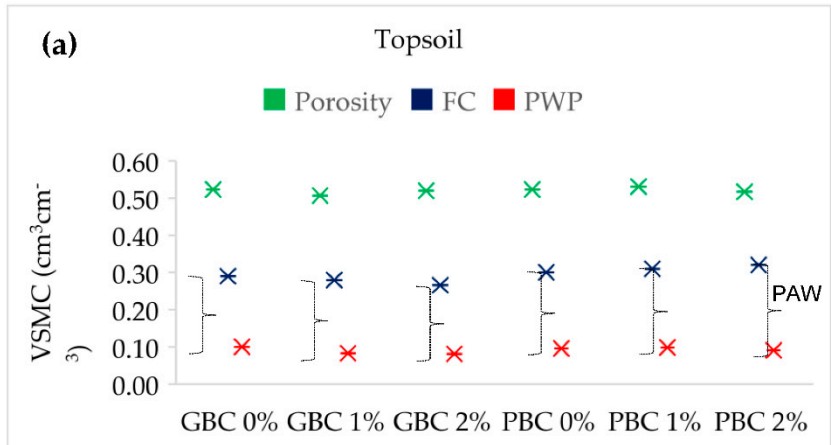

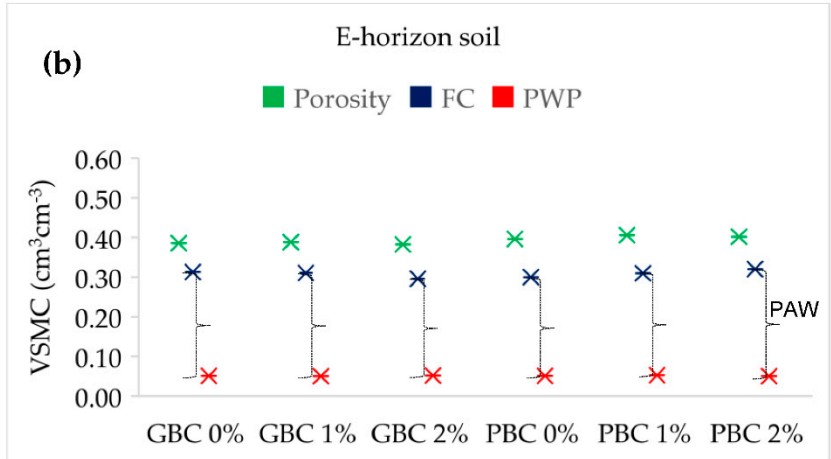

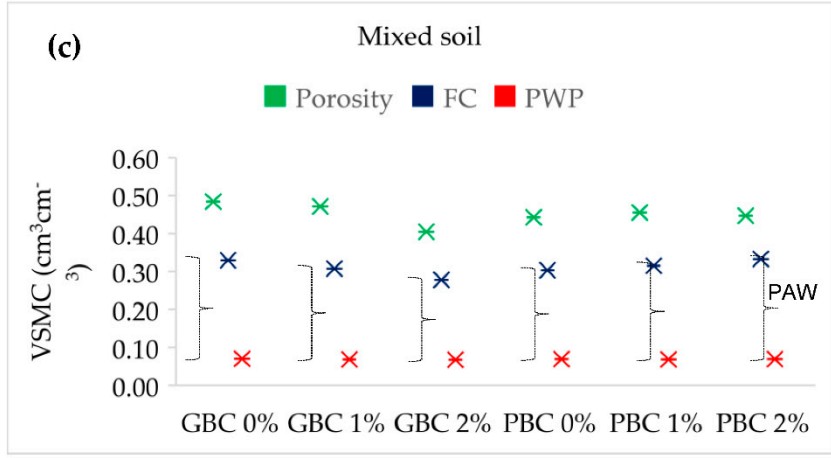

**Figure 5.** Effect of biochar amendment on porosity, field capacity (FC), permanent wilting point (PWP) and plant available water (PAW) in different types of soil: (**a**) topsoil; (**b**) E-horizon soil and (**c**) mixed soil. Abbreviations: GBC—granular biochar; PBC—powder biochar; VSMC—volumetric soil moisture content; the symbol "}" indicated PAW which is the difference between FC and PWP.

Table 3 summarizes van Genucthen (1980) [61] parameters and residual water content. All SWRC fitted parameters had a very high coefficient of determination ($R^2$) and were close to the unit

(Figure 4 and Table 3) that showed the accuracy of the VG model for predicting the SWRC in the soil amended with granular and powder biochar. This indicates that the fitting of the proposed interpolation equation to the experimental data was highly acceptable. Saturated and residual water contents of the treatment combination influenced by the types of biochar. For example, the powder biochar amendment influenced an increase in saturated and residual water content, while granular biochar induced a decrease in VSMC in the biochar-treated soil.

The 1:1 comparison of the measured and predicted values of VSMC for all the combinations are shown in Figure 6. Both measured and predicted values of porosity found in 1:1 line with $R^2 = 1$, and the values were statistically significant ($p = 0.000$). For FC, both measured and predicted values were found to be very close to the 1:1 line with a high value of $R^2 = 0.9962$, and the values were statistically significant ($p = 0.012$). VSMC at $-700$ kPa, both measured and predicted values lie in 1:1 line with high $R^2 = 0.9699$ and the values found statistically significant ($p = 0.002$). The results of all three VSMC parameters found a good fit in the 1:1 line.

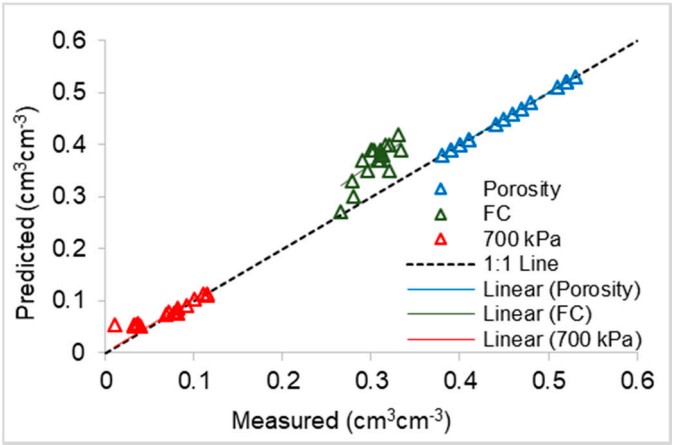

**Figure 6.** 1:1 Line graph of measured and predicted values of the porosity, field capacity (FC) and water content at $-700$ kPa. Abbreviation: VSMC—volumetric soil moisture content.

### 3.8.2. Plant Available Water (PAW)

PWP values of all the treatment combinations were calculated using predicted values from the van Genucthen (1980) model [61]. After determining the PWP values, we found PAW for each treatment combination. In the powder biochar-amended topsoil, PAW increased by 3.7% (with 1% biochar rate) and by 12.5% (with 2% biochar rate) when compared to the control topsoil. In contrast, PAW decreased by 2.9% in the topsoil with a 2% granular biochar amendment. In the case of powder biochar-amended E-horizon soil, PAW increased by 3.4% (with 1% biochar rate) and by 8.4% (with 2% biochar rate) when compared to the control. When we applied granular biochar in the same E-horizon soil, PAW decreased very slightly (0.6%) with a 1% biochar rate and decreased by 6.3% with a 2% biochar rate. In the powder biochar-treated mixed soil, PAW increased by 5.8% (with 1% biochar rate) and by 12.9% (with 2% biochar rate). On the other hand, in the same mixed soil, adding 1% granular biochar decreased PAW by 7.8% while adding 2% granular biochar decreased PAW by 19% when compared to the control mixed soil (Figure 5). These results suggest that granular biochar would not be suitable for increasing PAW, and the powder biochar can be applied to increase PAW at the crop root zone of the boreal podzolic soil.

### 3.9. Temporal Effects of Treatments on Soil pH

The temporal effects of biochar (from day 1 to day 7) on pH are shown in Figure 7. In the topsoil, granular biochar amendment increased pH from 5.7 to 6.0 (with 0.5%), from 5.8 to 6.0 (with 1%), and from 5.8 to 6.0 (with 2%) biochar rates from day 1 to day 7 where control topsoil pH was found to

be 5.3 at day 1. There was a slightly increasing trend observed in the temporal effect of pH amended with granular biochar in the topsoil (Figure 7a). Similarly, a slightly increasing trend was found when powder biochar was applied to the topsoil. However, E-horizon soil pH was found to be very low (pH = 4.3) in the control of E-horizon soil. After applying powder biochar to the E-horizon soil, pH increased from 4.3 to 5.5 from day 1 to day 7 with the biochar rate of 0.5 to 2% (Figure 7c,d). Increasing trends of pH were observed with the increasing rate of both granular and powder biochar in the E-horizon soil. Also, Figure 7e,f indicates a slightly increasing trend of pH in the mixed soil when both granular and powder biochar were applied at different rates. The initial pH of the mixed soil was 5.6, and after seven days, the pH increased to 6.1 (with 2% granular) and 6.3 (with 2% powder) after biochar application.

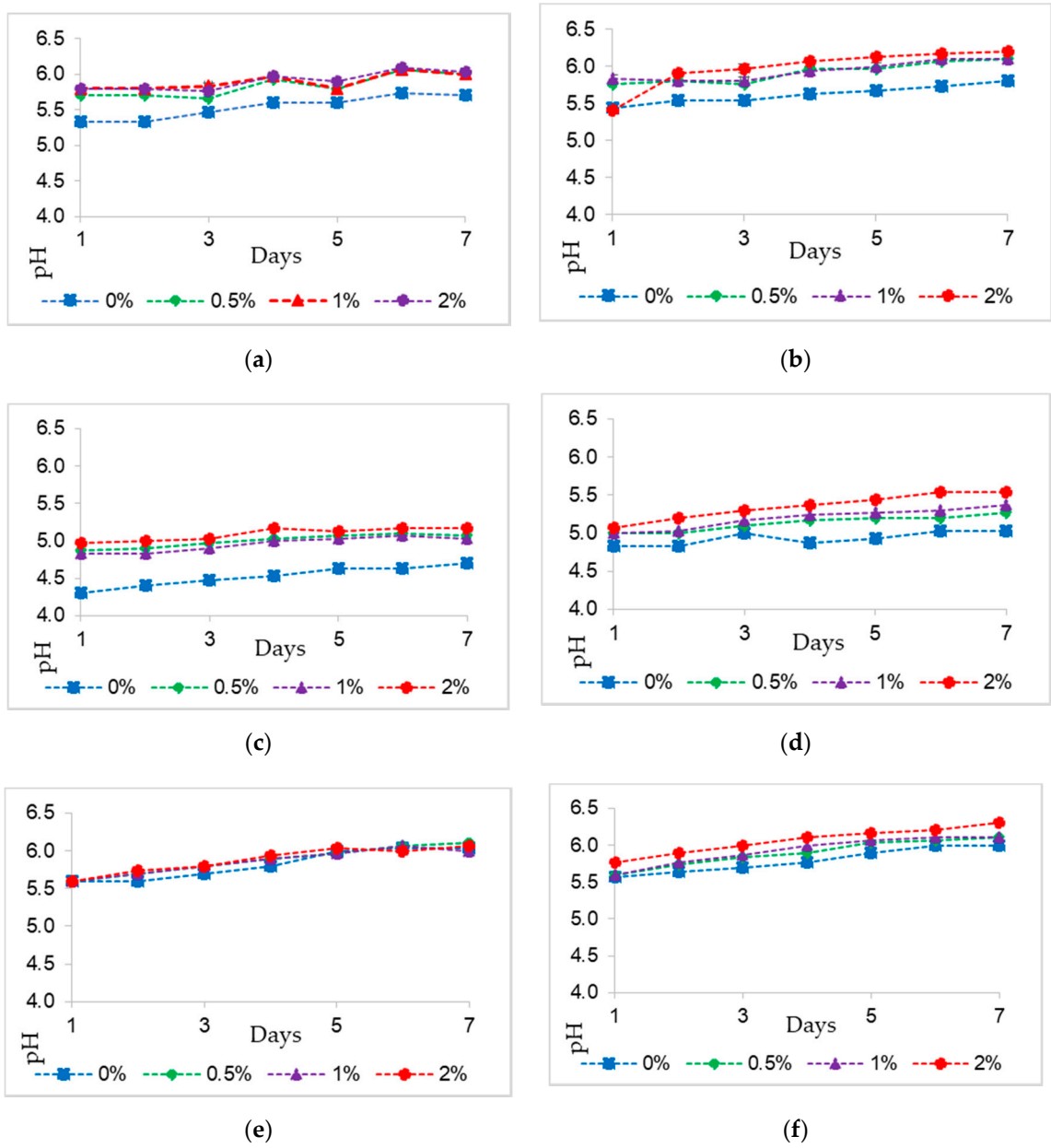

**Figure 7.** Temporal effect of biochar on pH: (**a**) topsoil with granular biochar; (**b**) topsoil with powder biochar; (**c**) E-horizon soil with granular biochar; (**d**) E-horizon soil with powder biochar; (**e**) mixed soil with granular biochar; (**f**) mixed soil with powder biochar.

## 3.10. Temporal Effect on Electrical Conductivity (EC)

Figure 8 shows the temporal effect of different types and rates of biochar application on different types of soils from day 1 to day 7. In the topsoil, granular biochar amendment increased EC from 0.0 to 0.13 dS/m, while powder biochar increased EC from 0.0 to 0.20 dS/m from day 1 to day 7 with 0.5 to 2% biochar rates. Factually, there was no impact of biochar observed on EC in the E-horizon soil. In the mixed soil, EC increased from 0.0 to 0.10 dS/m (with granular biochar) and from 0.0 to 0.17 dS/m (with powder biochar) when applying biochar at different rates from day 1 to day 7.

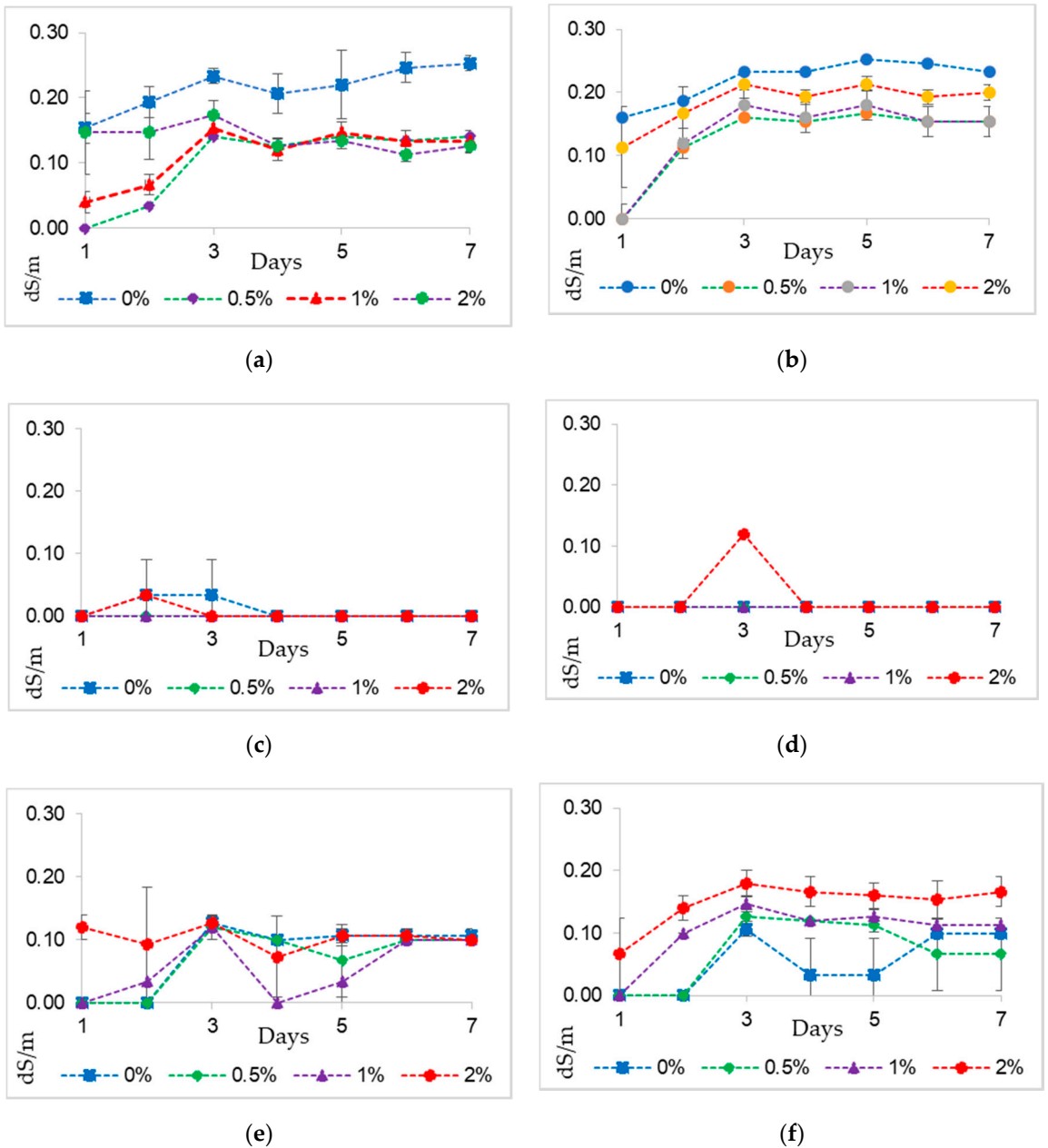

**Figure 8.** Temporal effect of biochar on electrical conductivity (EC): (**a**) topsoil with granular biochar; (**b**) topsoil with powder biochar; (**c**) E-horizon soil with granular biochar; (**d**) E-horizon soil with powder biochar; (**e**) mixed soil with granular biochar; (**f**) mixed soil with powder biochar.

## 3.11. Temporal Effect on Water Repellency vs. Moisture Content

Figure 9 showed the temporal effect of granular and powder biochar on moisture content (MC) vs. WR. The effect of biochar application on WR and MC was observed at FC level from day 1 to day 6. In the biochar (both granular and powder) treated topsoil, MC instantly dropped from FC on day 1. After a declining trend of MC and WR was observed from day 1 to day 3, the trend was found to be almost flat from day 4 to day 6. In the case of biochar-treated (both granular and powder) E-horizon soil, a huge drop of MC and WR was observed from FC level to after day 1. The slope of the curve was found to be almost flat from day 1 to day 6. In the case of both types of biochar-amended mixed soil, a continuous declining trend was observed in both MC and WR. After 6 days drying at 28 °C in the oven, the moisture content (maximum 3.35% and minimum 0.42%—described in Figure 10) was found to be below the permanent wilting point (around 5–9%). Overall, MC and WR showed declining trends from day 1 to day 6 with different types and rates of biochar amendment in all types of soil (Figure 10).

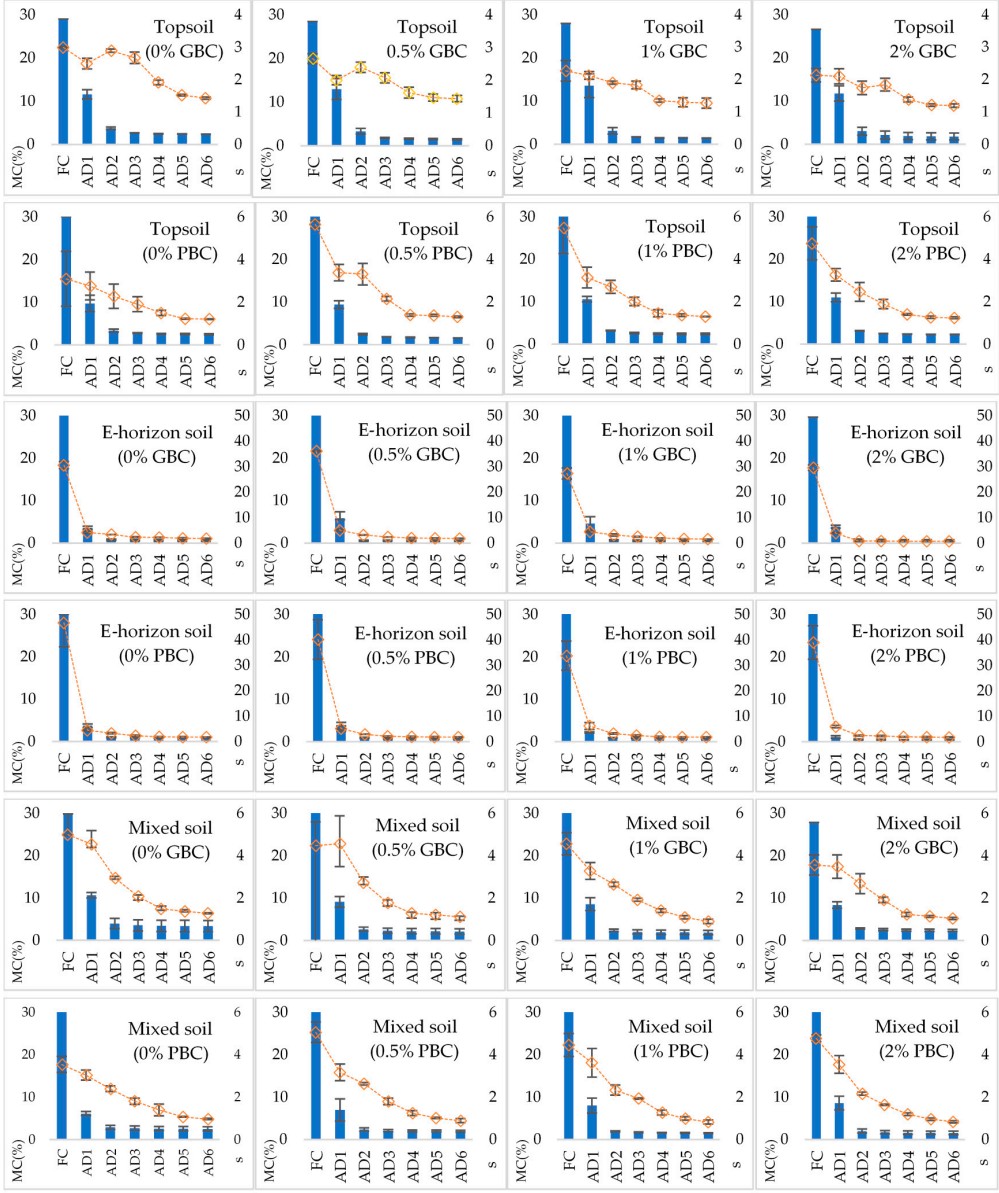

**Figure 9.** Temporal effect on moisture content (MC) vs. water repellency (WR) in biochar-amended soil. Samples were placed 28 °C in the oven after field capacity level from day 1 to day 6. Abbreviations: GBC—granular biochar; PBC—powder biochar.

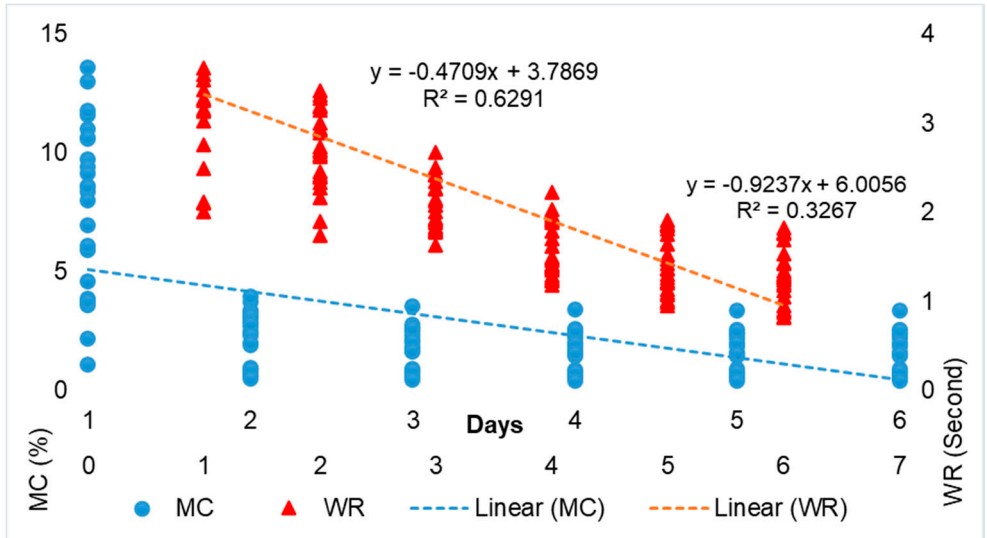

**Figure 10.** Combined temporal effect on moisture content (MC) vs. water repellency (WR) in biochar-amended soil from day 1 to day 6.

### 3.12. Principal Component Analysis

Principal component analysis (PCA) showed the relationship of shared physicochemical properties of three types of soil that were amended with two types of biochar with different rates (Figure 11a,b). PCA explained 72.20% of the total variability in the data set, where component 1 displayed 50.81% and component 2 displayed 21.37%, respectively, for variability. We observed different physicochemical properties in different quadrants of the PCA biplot. Two biochar types were grouped in different quadrants based on different physicochemical properties of the three types of soil found in the PCA observation plot (Figure 11a). A clear grouping of three types of soil was observed due to different variables under the study (Figure 11b). Topsoil and E-horizon soils were observed in opposite quadrants, whereas mixed soil is noted in between topsoil and E-horizon soil, which is expected due to the mixing of both types of soils (Figure 11b). The distribution pattern of parameters in different quadrants explained their associations, confirming strong positive or negative correlations between the parameters. The PCA results determined that strong and positive correlations exist among chemical properties such as EC, pH, CEC, TC, and all the parameters found in quadrant A. In the case of physical properties, strong positive relationships were observed between FC and PAW (exist in the same quadrant D). In addition, positive relationships were found among BD, FC, and PAW. However, strong negative relationships were observed between porosity and WR; BD and EC; TC and N; and CEC and N, as all relational data were found in the exact opposite quadrants. PCA delineated that variables of quadrant A were moderately and strongly connected with quadrant D. The findings from PCA suggest that different physicochemical properties, such as EC, pH, CEC, TC, and BD, FC, and PAW, had a strong relationship, while other parameters such as porosity and WR, and CEC and N had no relationship after applying biochar in podzolic soil in the boreal ecosystem.

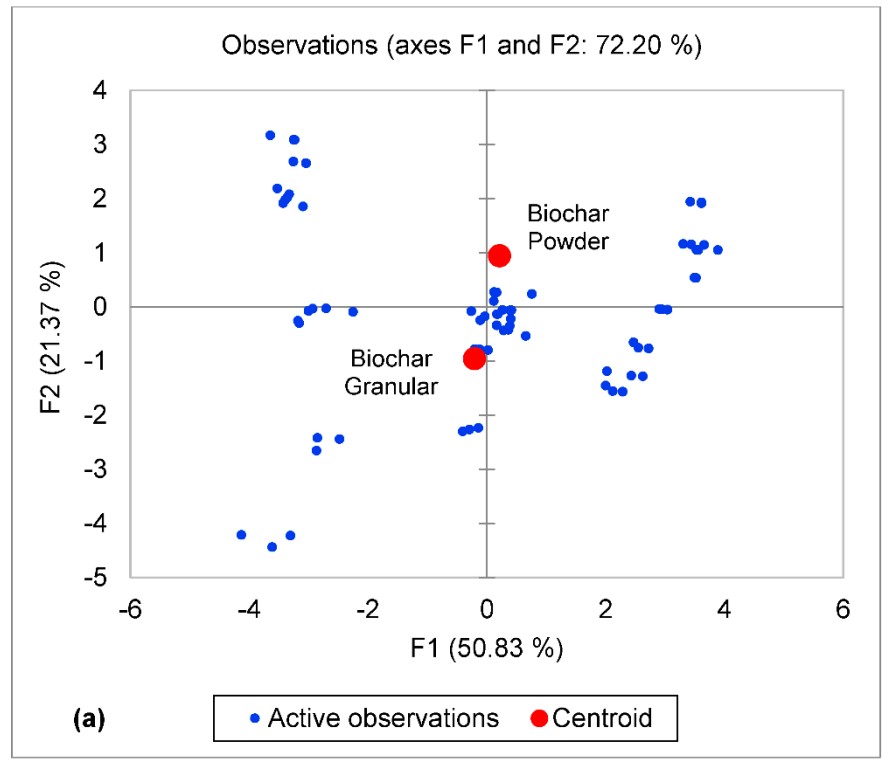

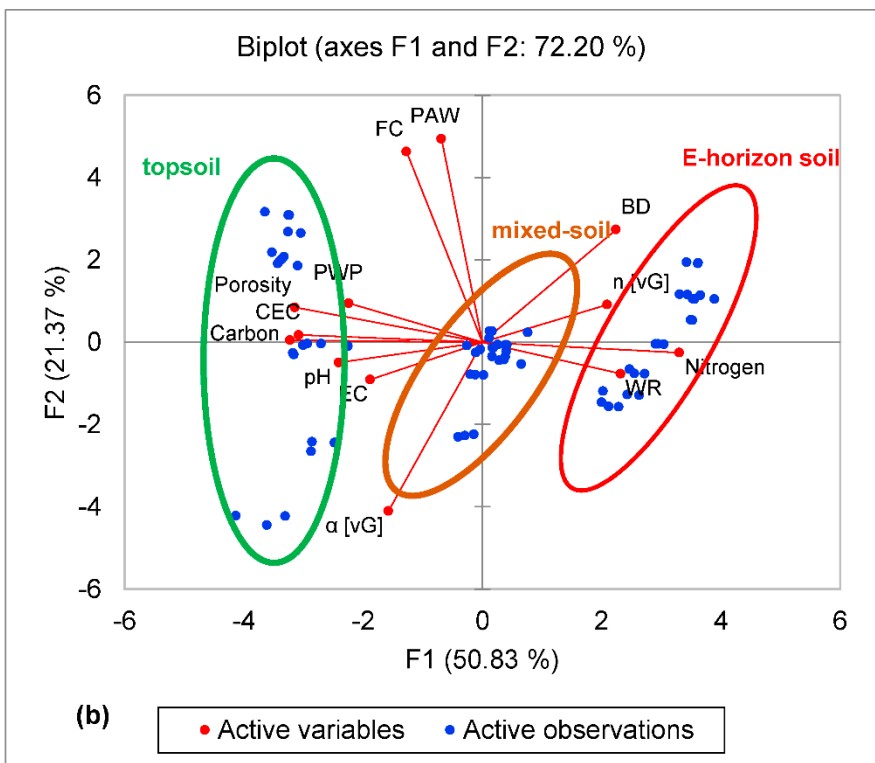

**Figure 11.** Principal component analysis (PCA) output of (**a**) observation and (**b**) biplot showing the effects of biochar on physicochemical parameters of podzolic soil.

## 4. Discussion

The study findings indicate that the biochar amendment significantly affects the physicochemical properties of the top, E-horizon, and mixed soil. The physicochemical properties of biochar-amended soil were correlated with the biochar types and rates. A higher percentage of biochar rates showed lower BD and increased hydraulic properties like porosity, FC, and PAW, especially when powder biochar applied to the top, E-horizon, and mixed soil. In 2019, Fu et al. [65] also found similar effects for biochar, suggesting that it significantly increased total porosity and decreased BD in biochar-amended soil. The changes in soil properties could be ascribed to the changes in soil structure. The effect of biochar on soil properties varied based on soil types. The findings from experiments signaled that biochar application in the mixed soil showed better consistency in improving physicochemical properties with respect to biochar rates compared to the biochar topsoil and E-horizon soil.

Biochar characteristic is highly influential on biochar function in the biochar-amended soil [2,66]. The pore size in biochar is considered one of the important characteristics that can increase PAW and improve nutrient retention in the soil. Combined biochar pore size and SA help to describe biochar capacity for adsorption of nutrients and retention of heavy metals. As water attraction of biochar is influenced by its SA, pore size and distribution pattern on the surface, it can be assumed that powder biochar would be more likely to possess hydrophilic characteristics (attracting more water in the soil), while granular biochar would be more likely to possess hydrophobic characteristics (preventing water release and entrance in the pores). The physical structure of biochar (e.g., pore size, SA, and particle morphology) is an important factor when changing soil physicochemical properties [67]. The morphological characteristics of both types of biochar, such as surface roughness and pore radius, could have a greater influence on the hydrological properties of biochar-treated soil. If the biochar possesses hydrophilic (wetting characteristics) or hydrophobic characteristics, the treatment combination will show that type of properties. Mandal et al. [55] mentioned that biochar usually has a microporous structure with higher pore volume and defined pore diameter with the surface area of 576.1 $m^2g^{-1}$. In this study, we found that granular biochar did not have any SA that was non-porous, while the SA of powder biochar was 12 $m^2g^{-1}$. The pore arrangement on the biochar surface was visible in the SEM image and BET analysis. Identified SA was located on the surface and inside of the biochar particles. Pore size and distribution pattern of granular and powder biochar were visible in the SEM image, which identified SSA for both types of biochar. The SEM image indicated that both granular and powder biochar share similar surface characteristics. Some important biochar properties, such as high total porosity, number of micropores, and SA, can potentially improve soil physical properties and create a better environment for plant root growth and nutrient uptake. Small size biochar particles, especially with diameter <0.5 μm, can reduce the volume of soil pores by filling the available larger pore spaces. On the other hand, larger biochar particles with a range of 0.5–500 μm can increase the total pore volumes by increasing macropores between particles and micropores within particles. It is suggested that SWR properties specifically depend on pore size, where water can be stored, and the size of the biochar particles [10].

Biochar could have hydrophilic and hydrophobic characteristics depending on the source materials and types. Hydrophilic biochar is usually providing superior wettability to soil [8]. Hydrophobic biochar surfaces may obstruct the uptake of water into biochar pores without considering pore size, distribution pattern, and structure. This may have a huge impact on WR (as WR depends on both biochar rates and soil particle size) that subsequently affects water infiltration in soil, SWR capacity as well as plant growth [16]. Rattanakam et al. [8] mentioned that hydrophobicity or hydrophilicity is not the most important governing factor of the SWR ability of soil.

Studies suggested that biochar pore sizes are classified as storage pores if the pore size ranges from 0.5 to 50 μm, which helps to increase water retention capabilities—thus increasing SWR, PAW, and nutrients retention in the soil [68]. However, we need to check the biochar pore distribution pattern on different sides and at different angles. If the percentage of the pore with the range 0.5 to 50 μm is found to be high compared to non-porous size, then the biochar could show hydrophilic

characteristics. The pore size distribution of biochar may influence water storage capacity and pore sizes range from 1 to 10 μm are defined as macro-porosity. The study by Hardie et al. [10] found that almost all (95%) pore size of the applied biochar was <0.002 μm. Studies indicated that smaller size pore diameter (<0.2 μm) biochar is expected not to alter water storage capacity significantly in the sandy soil because smaller pore size of biochar has a very little contribution in variation with regards to particle size distribution of the biochar soil mixture [33]. However, many factors influence the SWR characteristics of biochar-amended soil. The biochar application rate is recognized as one of the most important factors that affects hydraulic properties [15]. Biochar amendment in soil could have both direct and indirect impacts on SWR in the soil. Biochar amendment can directly impact the SWR in the soil due to its ability to create a larger surface area and a higher number of residual pores on the biochar where water can be retained through capillarity. The direct impact of biochar application in the agricultural field are improve total porosity in soil, increase MC, decrease water mobility in soil, and reduce water stress in plants' root zone. The experiment results indicated that application of powder biochar in three types of soil improve PAW that is a direct positive impact of biochar amendment to improve water stress condition. Fine biochar particles could directly contribute to increasing PAW by increasing porosity, generating more accommodation of pores, or by improving aggregate stability in the biochar-treated soil [10,33]. Biochar application in sandy soil can increase SWR and PAW. However, in the clay soil, both SWR and PAW properties may be reduced through biochar application. Besides, the indirect impact of biochar in soil includes improvements of aggregate stability and structural composition in the soil, and consequently affecting the capacity of SWR in the biochar-amended soil [13,69].

Specifically, soil amended with higher porosity biochar has been shown to improve SWR [15]. Our data suggest that biochar with larger pore volume and average pore diameter has better SWR capacity. Rattanakam et al. [8] reported that the biochar amendment on sandy soil increased SWR by 15.9% when compared to the control. More water can be retained in sandy soil when biochar is applied in a single layer [70]. If biochar is mixed non-uniformly within the soil, then the chance of high SWR capacity could be disrupted. In the experiment, the soil amended with granular biochar, porosity showed a slight decline with increasing biochar rates compared to powder biochar. Usually, granular biochar shape is not uniform and when added in soil it can creates more void spaces and increase the porosity. However, SEM image showed that some of the granular biochar walls were smooth with uniform angles in some instances. Additionally, the granular biochar showed hydrophobic characteristics. These could be the potential reasons why granular biochar-treated soil showed slightly decreasing trend in porosity. Biochar particle size has a significant influence on the hydraulic properties of biochar-treated soil. Biochar amendment in agricultural soil maximizes the water use efficiency, even in sandy loam and sandy soil, thus enhancing agricultural production [29].

Biochar amendment in agricultural soil may be one of the most suitable and sustainable options to provide long-lasting improvements in soil fertility, especially in sandy soils where agricultural practices face constraints due to low SWR and its tendency to leach high amounts of soil nutrients. Jeffery et al. [16] and Villagra-Mendozaa and Horn [33] mentioned that biochar properties such as SSA and total pore volume, play an important role in WHC capacity. In the study, we found that powder biochar application increased field capacity the treated soil compared to control soil. Thus, the hydraulic properties (WHC) improved in the biochar-amended podzolic soil. But application of granular biochar in the soil showed slightly decreased WHC compared to control. The results confirm that the applied powder biochar could possess hydrophilic characteristics while granular biochar showed hydrophobic properties.

Results from our experiments showed that both types of biochar have a significant influence on soil properties, including increasing pH, EC, CEC, TC: N ratio, WR, porosity, FC, and PAW and decreasing BD when biochar dosage increases. In the biochar-treated podzolic soil, the improvement of hydrological properties could be attributed to the micro-porosity in powder biochar and macro-porosity in the granular biochar. Biochar addition enhances the physical and hydraulic properties of soil

because it is a porous substance [33]. Even though biochar increased porosity in the soil, SWR capacity and PAW would sometimes remain unchanged [10]. Biochar-amended soil posits strong SWR 28–32% in the sandy loam soil [15] and secures available nutrients in the soil that may enhance sandy soil quality [71,72]. Głąb et al. [73] indicated that the application of the finest biochar particles in sandy soil increased PAW and WR slightly, even though it was classified as non-repellent. In the study, WDPT test also confirmed that all the treatment combinations were classified as slightly-water repellent.

Besides improving physical properties of soil, biochar amendment usually improves the chemical properties of soil. In 2016, Gamage et al. [74] found significant changes in soil physicochemical properties at 1% with 0.5% biochar application rates. Different studies across the world found similar effects for the biochar amendment on the soil properties in their region. For example, biochar application showed potential ameliorating acidic sandy soils by increasing pH and CEC, especially for sandy loam soil [74]. In the E-horizon soil (sandy soil), pH of the control soil was 4.3. After adding 2% powder biochar, pH increased up to 5.5 in the treated E-horizon soil. Besides, in granular biochar-treated E-horizon soil and mixed soil, CEC showed a slightly decreasing trend compared to the powdered biochar treatment. The CEC of powdered biochar ($5.76 \pm 0.31$ cmolkg$^{-1}$) was found to be low compared to granular biochar ($11.07 \pm 0.70$ cmolkg$^{-1}$) resulting in lower CEC values in soils treated with powdered biochar. Besides, the mixed soil consists of topsoil and E-horizon soils with a ratio of 2:1. CEC of E-horizon was found to be very low ($2.61 \pm 0.27$ cmolkg$^{-1}$). This could be another reason why the CEC value of powder biochar-treated mixed soil was low compared to granular biochar-treated mixed soil. Additionally, in the experiment, we found no significant changes on EC in the E-horizon soil (which was a sandy loam soil). Gamage et al. [74] indicated that there was no significant impact on EC in the biochar-amended sandy loam soils. There was no significant increase in EC in either soil or biochar treatment that indicates no threat of salinity in the boreal podzolic soil.

We applied biochar containing 60% of TC, which was close to the percentage used in the Laird et al. [75] study where they used biochar with 71.5% of TC. Study findings confirmed that the biochar amendment in soil exhibited a prominent effect on soil TC through an increased C:N ratio. In 2019, Majumder et al. [76] indicated that biochar increased the amount of C in the treated soil. In addition, Laghari et al. [17] found that higher doses of biochar improved C:N ratio in sandy desert soils. They recommended biochar as a suitable option for desert soil application to aid in sustainable agricultural development. Thus, the biochar amendment in soil was found to be beneficial with C stability, N cycling, and the addition of N fertilizer as well [77]. The study results we found in PCA analysis highlighted several interesting interrelationships among the physicochemical properties of soil after being amended with biochar. Additionally, in PCA analysis, different properties provided consistent results with the cluster wise analysis of the properties [3].

In this study, the biochar amendment in the three types of soil had a significant impact on the alteration of several physicochemical soil properties—the similar impact of biochar application in soil was observed by Arbestain et al. [78]. Ojeda et al. [34] mentioned that three important soil functions, nutrient release, water, and carbon storage are influenced by the biochar amendment. Biochar plays an important role in binding material to form stable micro-aggregates in the soil. The micro-aggregates facilitate the formation of capillaries in the treated soil and thereby increased SWR properties [29]. When we added biochar to the soil, organic matter (humic substances), microorganisms, heavy metals, etc., were all expected to occupy the volumes of unoccupied pores [13]. Biochar micropores generally play a vital role in increasing the soil matrix micropores and increasing the diameter of the pore space (macropores), which could lead to increased earthworm burrowing—as an assumption by Hardie et al. [10]. Thus, biochar pore size and distribution patterns are considered crucial factors for the formation of hydraulic properties in the biochar-treated soil, especially in sandy soil [70] and sandy loam soil [20,29]. In our experiment, we found biochar application to be beneficial on loam and sandy loam podzolic soil in the boreal ecosystem. The experiment confirmed that the powder biochar amendment in soil, especially with 2% rate, more efficient in improving availability of PAW. This could be able to improve water stress situation and enhance soil–water–plant interaction.

In a study, Suliman et al. [28] also mentioned that biochar application in podzolic soil could be considered a sustainable option for improving soil fertility for the long-term, especially in sandy soil. Study findings from Villagra-Mendozaa and Horn [33] recommended that the biochar amendment in sandy soil is beneficial for soil health, as well as plant growth, especially in climate-sensitive (cool) conditions. These findings suggest that biochar is proven to be a long-lasting sustainable option as a soil amendment.

Based on the experimental findings and discussion, future points to be considered are:

- Hydrological functions of biochar-amended soil need to be investigated at different environmental conditions (field experiments), which will help to predict potential effects of different types and rates of biochar, especially higher doses on SWR. In addition, the agronomic benefits of biochar application in the soil need to be evaluated [79].
- The physicochemical properties of biochar may change after a period of environmental exposure. For example, porosity (surface structural characteristics) is likely to be gradually altered by 4 years of exposure [36,80]. This can affect the properties of biochar-amended soil and make it a challenge to predict ecosystem services in the long-term. Thus, the long-term stability of biochar in soil and its impact on soil physicochemical properties need to be investigated.
- Quantitative risk assessment of biochar application on human health and ecosystem [81] is very important to ensure proper safety, development, and sustainability if biochar application use is to increase in boreal agricultural system.

## 5. Conclusions

The experiment results indicated that the biochar amendment significantly improved physicochemical properties of podzolic soil. The basic physical properties of biochar-treated soils, such as porosity and field capacity, were found to be dependent on biochar rate, morphology, and granulated/powdered structure. Granular biochar was determined to be slightly hydrophobic, and powder biochar was determined to exhibit hydrophilic characteristics. This study provided strong evidence that supports the positive impact of the biochar amendment in podzolic soil. For example, biochar application increased pH (especially in the E-horizon soil), electrical conductivity, cation exchange capacity, and the total carbon: nitrogen ratio in the soil while reducing bulk density. In this study, water repellency was found to be slightly increased with the different rates of biochar treatment, but the treatment combinations were still classified as slightly-repellent. The biochar amendment showed great ability in improving soil water retention. Both types of biochar amendments increased porosity in the treatment combinations. However, the field capacity and plant available water increased more in the powder biochar-amended soil than the granular biochar-amended soil. The study results suggested that the application of 2% powder biochar could have a beneficial influence on the physicochemical properties of podzolic soil. The overall experimental findings indicated that the application of biochar in mixed soil showed better improvement in physicochemical properties in response to increasing biochar rates, compared to the improvements found in its application to the topsoil and E-horizon soil. Further chemical and structural analyses are needed to accurately calculate the percentage of pore space changed through biochar application. Biochar application as soil amendments in podzols needs to be investigated more to better understand its effects in improving soil physicochemical properties in boreal ecosystem. The long-term ecotoxicological effects of biochar need to also be examined to ensure ecological sustainability in the future.

**Author Contributions:** R.S.: Identification of the research topic, study design, experiment execution, data analysis, presentation of results and original draft manuscript preparation; L.G.: Identification of the research topic, study design, resources, review, editing and supervision; R.T.: study design, review, and editing; M.N.: experiment execution, data analysis, and manuscript review; K.H.: review and editing. All authors have read and agreed to the published version of the manuscript.

**Funding:** This research was financially supported by the School of Science and the Environment, Grenfell Campus, and Seed-Bridge-Multidisciplinary grant of Memorial University of Newfoundland, Canada.

**Acknowledgments:** Authors acknowledge members of Hydrology & Agrophysics Group, Grenfell Campus-Memorial University of Newfoundland for their help and constructive comments during the study. Authors gratefully acknowledge Tao Yuan, Lab Coordinator of the Boreal Ecosystem and Research Facility, Grenfell Campus, Memorial University of Newfoundland, for his valuable contribution in conducting analysis, especially on Ion Chromatography, and overall coordinating activities in the lab. Special thanks go to Muhammad M. Farhain and Gayantha S. Perera for their help during soil sampling and sample preparation. Authors are also thankful to Yeukai Katanda, Postdoctoral Fellow, School of Science and the Environment, Grenfell Campus, Memorial University of Newfoundland, for her support given during the experiments, especially on methodology and execution of the experiments. Authors also acknowledge Gabriel Grunwald, Department of Naval Architecture, Marine Institute of Memorial University of Newfoundland, Canada for copy-editing of this manuscript.

**Conflicts of Interest:** The authors declare no conflict of interest in this experiment-based study. Comprehensive study design was prepared and executed without having any influences. The findings presented in this article were obtained through our experiments. The findings were also compared with the published peer-reviewed journal articles' findings.

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
