# Peer review of "Investigating the Influence of Biochar Amendment on the Physicochemical Properties of Podzolic Soil"

_agriculture, doi:10.3390/agriculture10100471_

Round 1
Reviewer 1 Report
A report for: Investigating the influence of biochar amendment on the physicochemical properties of podzolic soil.
It is an interesting paper that aimed to investigate the effects of biochar on the physicochemical properties of podzolic soils.
The paper demonstrates the novelty and importance of biochar amended that reach the significance expected from one original scientific paper. The work presented here is of relevance to other studies or how to extend methodological approaches beyond the state-of-the-art on this topic.
To the best of my knowledge the experimental labor has been carried out with care and the conclusions have been inferred prudently. Nevertheless, some paragraphs could be improved as suggested below.
Line 105-119. Explain better the objectives.
Line 128-129 Three soil types were evaluated: topsoil, E-horizon soil, and mixed soil (2:1 Topsoil and E-horizon soil ratio according to the average soil layers in the field). Explain better.
Line 151-253 The methodology is too extensive, please reduce the length.
Line 306-308. I do not understand how all soil mixtures will have the same particle distribution. Explain it.
Line 345-352 The Total Carbon values I don't think are good. Review the results.
Line 590 clarify the sentence.
Line 688-705, please reduce it.
Author Response
Response to Reviewer 1 comments_ agriculture-913735
General comments: It is an interesting paper that aimed to investigate the effects of biochar on the physicochemical properties of podzolic soils. The paper demonstrates the novelty and importance of biochar amended that reach the significance expected from one original scientific paper. The work presented here is of relevance to other studies or how to extend methodological approaches beyond the state-of-the-art on this topic. To the best of my knowledge the experimental labor has been carried out with care and the conclusions have been inferred prudently. Nevertheless, some paragraphs could be improved as suggested below.
Reply: We thank the reviewer for the positive and constructive comments. We have addressed all specific comments, and these are highlighted in blue font color. We hope that the reviewer will find our response satisfactory to recommend the manuscript for publication.
Comment 1: Line 105-119. Explain better the objectives.
Reply: We thank the reviewer for highlighting this important point. We have revised the text with better explanation of study objectives as “The objective of the study was to investigate the influence of biochar amendment on the physicochemical properties of podzolic soil. Overall, the goal of the study was to evaluate the effects of different types of biochar on important soil physicochemical properties that can influence crop performance when cultivated on podzolic soils in boreal ecosystem”. Please see page 3 line 113 – 117 for such revisions in the manuscript.
Comment 2: Line 128-129 Three soil types were evaluated: topsoil, E-horizon soil, and mixed soil (2:1 Topsoil and E-horizon soil ratio according to the average soil layers in the field). Explain better.
Reply: We thank the reviewer for good suggestions to improve the quality and readability of our manuscript. We included the following text in the revised manuscript for more clarity: “During soil sampling from the field, we observed that the average depth of topsoil was around 10 cm, while the below E-horizon soil was around 5 cm thick. Before cultivation, soil is usually ploughed forming Ap horizon consisting a mixed topsoil and E-horizon. Based on these observations the mixed soil was prepared combing topsoil and E-horizon soils with a ratio of 2 (topsoil): 1 (E-horizon soil) - according to the average soil layers in the field. Three soil types were evaluated: topsoil, E-horizon soil, and the mixed soil.” Please see page 3 line 134 - 139 for said changes in the revised manuscript.
Comment 3: Line 151-253 The methodology is too extensive, please reduce the length.
Reply: Considering reviewer 1, 2 and 3 comments, we have revised and improved the methodology section. We kept the method section sufficiently detailed so the results can be reproducible in any event where readers of the journal would want to conduct similar experiments or try to replicate our study. We hope the reviewer will find the revised changes satisfactory for recommending acceptance of the manuscript for publication.
Comment 4: Line 306-308. I do not understand how all soil mixtures will have the same particle distribution. Explain it.
Reply: We thank the reviewer for this inciteful comment. We measured and calculated the particle size distribution of the three soil types used in this experiment (topsoil, E-horizon soil, and mixed soil). Sand content of topsoil = 60% and E-horizon = 58% and the calculated sand content of the mixed soil = 59% at 2:1 mixing ratio (60+60+58/3 = 59). This clarification is now added in the manuscript “and calculated the particle size in the mixed soil according to the mixing ratio of topsoil (2): E-horizon soil (1)” on page 4 line 164.
Comment 5: Line 345-352 The Total Carbon values I don't think are good. Review the results.
Reply: As suggested, we checked the total carbon values in all mixtures representing the three types of soils and two types of biochar. We added biochar at 0.5, 1 and 2% in the mixture and calculated percentage of total carbon of each biochar treated mixture based on the percentage of total carbon measured in the soil samples and biochar. Relevant study findings are described in the discussion section in page 24 line 684 to 692.
Comment 6: Line 590 clarify the sentence.
Reply: We thank the reviewer for highlighting this issue. The sentence is rephrased for more clarity as “Studies suggested that biochar pore sizes are classified as storage pores if the pore size ranges from 0.5 to 50 µm, which helps to increase water retention capabilities – thus increasing SWR, PAW, and nutrients retention in the soil [81]”. Please see page 21 line 605- 607 for these changes in the revised manuscript.
Comment 7: Line 688-705, please reduce it.
Reply: We agree with the reviewer’s thoughts and deleted the less important point as suggested. Please see page 23 line 722 reduced in the revised manuscript.
“Local environmental and soil conditions are very important in acidic soil management in the boreal ecosystem. Thus, environmental conditions need to be assessed when using biochar as a soil amendment under field conditions in the boreal region”.

Reviewer 2 Report
Authors have presented the manuscript “Investigating the influence of biochar amendment on the physicochemical properties of podzolic soil” where the effects of different biochars on the physicochemical properties of podzolic soils where studied. Experimental treatments consisted of three types of soils (topsoil, E-horizon soil and mixed soil), two biochar types (granular and powder) and four biochar application rates (0%, 0.5%, 1% and 2% on a weight basis). Ten physicochemical parameters were evaluated (bulk density, porosity, field capacity, plant available water, water repellency, electrical conductivity, pH, cation exchange capacity, total carbon, and nitrogen).
The results of the manuscript are interesting and the manuscript is written with accuracy, so I recommend this manuscript to be published after minor revisions.
I ask authors to consider some additional aspects, that are:
1.In figure 2 and 3 I suggest to present SEM images with the same magnification, in order to compare easily granular and powder biochars (in figure 3 there is a formatting issue with labels d2 and f2).
2. Why when you increased the rate of biochar on the treatment combination, you observed that water repellency increased with the biochar application rate?
3. In table 2, considering the mixed soil, why there is a decrease of cation exchange capacity for powder biochar while for granular biochar a different trend is evident?
4. In figure 5a, how do you explain the decreasing in porosity from 0% till 1% of GBC and an increase in the same parameter from 1% till 2% of GBC?
5. Can you discuss more deeply the effect of electrical conductivity due to biochar addition to the soils? In figure 8d is the 3-days 2% powder biochar in E-horizon soil an oulier value of EC?
6. In the text, you assess “Hardie et al. [10] found that almost all (95%) of the pores of the applied biochar were <0.002 μm diameter, with pore sizes ranging from 1 to 10 μm (potential macro-porosity).” But if pore sizes are higher than 1 μm they cannot be smaller than 2 nm.
7. Considering formatting, in some parts of the manuscript there are sentences that are written in bold, please check.
Author Response
Response to Reviewer 2 comments_ agriculture-913735
General comments: Authors have presented the manuscript “Investigating the influence of biochar amendment on the physicochemical properties of podzolic soil” where the effects of different biochars on the physicochemical properties of podzolic soils where studied. Experimental treatments consisted of three types of soils (topsoil, E-horizon soil and mixed soil), two biochar types (granular and powder) and four biochar application rates (0%, 0.5%, 1% and 2% on a weight basis). Ten physicochemical parameters were evaluated (bulk density, porosity, field capacity, plant available water, water repellency, electrical conductivity, pH, cation exchange capacity, total carbon, and nitrogen). The results of the manuscript are interesting and the manuscript is written with accuracy, so I recommend this manuscript to be published after minor revisions.
Reply: We thank the reviewer for the positive and constructive comments on our manuscript to improve the quality. We have addressed all specific comments, and these are highlighted in blue font color.
Comment 1: In figure 2 and 3 I suggest to present SEM images with the same magnification, in order to compare easily granular and powder biochars (in figure 3 there is a formatting issue with labels d2 and f2).
Reply: We thank the reviewer for highlighting this important point. Accordingly, we now present the same magnification for SEM images in both granular (Fig. 2) and powder (Fig. 3) biochar. For granular biochar, we have high magnification images (19998 x mag and 49999 x mag), so that we presented these images in Figure 2 (2f & 2g). However, for the powdered biochar, we have images up to 10001 x mag. Therefore, we presented the available SEM images up to 10001 x mag. We changed Figure 2b and Figure 3d2. The new figures with appropriate label inserted in the figure 2 & 3. Also, we changed the description of the results according to the changed figure label. We also addressed the formatting issue in Figure 2 & 3. All the labels in Figure 2 & 3 were checked and revised according to these changes. Please see Figure 2 & 3 in page 9 & 10 for these changes in the revised manuscript.
Comment 2: Why when you increased the rate of biochar on the treatment combination, you observed that water repellency increased with the biochar application rate?
Reply: This is a very good question. We used two types of biochar: granular and powder. Granular biochar showed hydrophobic characteristics and powder biochar showed hydrophilic characteristics. That is why when we increase biochar rate in the treatment combination, the water repellency increased in the treatment combinations, especially when we applied granular biochar in the treated soil.
Comment 3: In table 2, considering the mixed soil, why there is a decrease of cation exchange capacity for powder biochar while for granular biochar a different trend is evident?
Reply: Thanks for this question. For the clarification, we added these sentences in the discussion section: “Besides, in granular biochar treated E-horizon soil and mixed soil, cation exchange capacity (CEC) showed slightly decreasing trend compared to the powdered biochar treatment. The CEC of powdered biochar (5.76±0.31 cmolkg−1) was found to be low compared to granular biochar (11.07±0.70 cmolkg−1) resulting in lower CEC values in soils treated with powdered biochar. Besides, the mixed soil consists of topsoil and E-horizon soils with a ratio of 2:1. CEC of E-horizon was found to be very low (2.61±0.27 cmolkg−1). This could be another reason why the CEC value of powder biochar treated mixed soil was low compared to granular biochar treated mixed soil”. Please see the page 23 line 673-680.
Comment 4: In figure 5a, how do you explain the decreasing in porosity from 0% till 1% of GBC and an increase in the same parameter from 1% till 2% of GBC?
Reply: We thank the reviewer for highlighting this important point. We have added these sentences in the discussion section: “In the experiment the soil amended with granular biochar, porosity showed a slight decline with increasing biochar rates compared to powder biochar. Usually, granular biochar shape is not uniform and when added in soil it can creates more void spaces and increase the porosity. However, SEM image showed that some of the granular biochar walls were smooth with uniform angles in some instances. Additionally, the granular biochar showed hydrophobic characteristics. These could be the potential reasons why granular biochar treated soil showed slightly decreasing trend in porosity”. Please find the added text in the page 22 line 636-642.
Comment 5: Can you discuss more deeply the effect of electrical conductivity due to biochar addition to the soils? In figure 8d is the 3-days 2% powder biochar in E-horizon soil an oulier value of EC?
Reply: We thank the reviewer for this important suggestion. We have included more text to highlight the effects of biochar on electrical conductivity (EC) in the soil in the discussion session on page 23 line 682-684 as “There was no significant increase of EC in either soil or biochar treatment that indicates no threat of salinity in the boreal podzolic soil.” In figure 8d, 2% powder biochar in E-horizon soil showed an outlier value of EC that could be an anomaly as there were almost no EC values observed in the biochar treated E-horizon soil.
Comment 6: In the text, you assess “Hardie et al. [10] found that almost all (95%) of the pores of the applied biochar were <0.002 μm diameter, with pore sizes ranging from 1 to 10 μm (potential macro-porosity).” But if pore sizes are higher than 1 μm they cannot be smaller than 2 nm.
Reply: We appreciate the reviewer’s insightful comment to improve the manuscript quality. We agree that if the pore sizes are higher than 1 μm, they cannot be smaller than 2 nm [1 nm = 1000 μm]. For the clarification, we have reprhased the above sentence as “The pore size distribution of biochar may influence water storage capacity and pore sizes range from 1 to 10 μm are defined as macro-porosity. The study by Hardie et al. [10] found that almost all (95%) pore size of the applied biochar was <0.002 μm”. Please see page 21 line 610-612 for the said changes in the revised manuscript.
Comment 7: Considering formatting, in some parts of the manuscript there are sentences that are written in bold, please check.
Reply: We thank the reviewer for highlighting the formatting errors in the manuscript. We have carefully revised the manuscript considering these suggestions. We hope the revised version is much improved and follow the Agriculture-MDPI author’s guidelines.

Reviewer 3 Report
Review of “Investigating the influence of biochar amendment on 3 the physicochemical properties of podzolic soil”
Abstract
L16 … biochar application on the…
L22 you have 3 soils x 2 biochar types x 4 applications x 3 repetitions = 72 experimental units. Do not include the analysis you made. Explain better the experimental design; is it a split plot with 3 repetitions (or blocks)? Did you do it in pots? How large?
L25. Do not include the data here (keep them for the results section), briefly explain which factor where significant and the main significant differences among treatments.
L31 the combinations are those from the mixed soil?
L33 explain better “the temporal effect”, did you take samples at day 1 and then at day 7?
L35 “…2% biochar application is the best combination…” of powder or granular biochar? For which soil?
Keywords: do not include those words already included in the title, like physicochemical properties and podzolic soil
Intro
L76 define “PAW”
L81-82: rephrase
L92 coarse?
M&M
In general, M&M should be described much briefly, without the equations (with the reference is enough). Only topsoil and E-horizon where analysed, not the “mixed soil”. For the “treatments combinations” values of BD, TC, N have been calculated, not measured.
I do not think the experimental design, the methodology and the statistics are adequate and rigorous for a research paper.
L122 the soil you took for the “agricultural experimental station”, was subjected to any management practices?. Tillage? what crops where in the experimental station grown? Fertilization?
L125 what was the soil classification beyond “podzol”, which horizons did it have, at which depths… explain better “Topsoil and E-horizon soil samples were collected from 0-15 cm depth”
Clima classification? Rainfall? Mean temperatures?
Table 1 belongs to the result section
L137 soils where amended in the lab? Why not in the field? Explain better the experimental design
Fig 1. you do not have 720 experimental units
L160 you did not analyse the “mixed soil”? and the values of sand, silt and clay from table 1?
L161 include the soil fractions sizes
L163 In what treatments was BD measured? Because in L167 you explain how you calculated BD for treatment combinations
L164 I don´t think the way you measured BD is rigorous “tapped three times on the table”??? why 3 and not 5? In the reference you gave they do not do it like that [51] (Agriculture 2019, 9(6), 133; https://doi.org/10.3390/agriculture9060133): “Bulk density was measured by collecting undisturbed soil core samples and dividing the dry soil mass by the core volume“
L175 water repellency was then estimated in soil and biochar samples but not in the combinations?
L183 I do not understand how conducting the WDPT at FC (I guess it is field capacity) and after 6 days drying at 28º in an oven (which will be close to permanent wilting point?) account to “measure temporal effects”
L190 “from each biochar treat soils” means 3 soils x 2 biochar x 4 doses x 3 replications?
L199 if only sodium concentration was measured is not the CEC, and the other cations like K, Ca, Mg, Al, Fe?
L212 how did you calculate C and N? sometimes adding biochar or other amendments into the soil does not result in a proportional increase of TC and N, how did you account for priming effects? Mineralization rates might change
L254 conducting one way anova you do not get the treatment interactions.
L258 explain better the control.
Results
Since most of the data for the "treatment combinations" have been calculated and not measured I do not think the results can be extrapolated to what would happen in real field conditions.
L313. You did not “observed” you calculated the theoretical BD for the “treatments combinations”
L315 of course you expected a high correlation between BD and biochar ratio since you have calculated BD proportionally to biochar rates
Discussion
Author Response
Response to Reviewer 3 comments_ agriculture-913735
We thank the reviewer for the positive and constructive comments. We have addressed all specific comments, and these are highlighted in blue font color. We hope that the reviewer will find our response satisfactory to recommend the manuscript for publication.
Abstract
Comment 1: L16 … biochar application on the…
Reply: We thank the reviewer for this suggestion and text is revised in the manuscript. Please see page 1 line 17 “application” for the said changes.
Comment 2: L22 you have 3 soils x 2 biochar types x 4 applications x 3 repetitions = 72 experimental units. Do not include the analysis you made. Explain better the experimental design; is it a split plot with 3 repetitions (or blocks)? Did you do it in pots? How large?
Reply: We thank the reviewer for this insightful comment. Experimental treatments included 3 soil types × 2 biochar types × 4 biochar rates × 3 replications making a total 72 experimental units. The experiment was a complete randomized design with factorial combinations.
- As suggested, we mentioned 72 experimental units (excluded 10 physicochemical parameters that analyzed in the study). we have corrected the number of experimental units in the text and figure as well. Please see the change in the page 1 line 23 and 4 line after 150.
- We collected soil samples from sampling site. After that, all the experiments were conducted in the laboratory. We followed the standard procedure for each experiment for evaluating the physicochemical properties presented in this manuscript.
Comment 3: L25. Do not include the data here (keep them for the results section), briefly explain which factor where significant and the main significant differences among treatments.
Reply: We agree with the reviewer’s suggestion and specific suggestion are followed in the revised manuscript. Please see page 1, line 26-28 for such changes made in the manuscript as indicated in blue font color: “The result indicated that the E-horizon soil was highly acidic compared to control (topsoil) and mixed soils. A significant difference was observed between the control and 2% biochar amendment in all three soil mixtures tested in this experiment”.
Comment 4: L31 the combinations are those from the mixed soil?
Reply: For more clarity and readability, we have revised the text as “all the biochar-soil combinations were classified as slightly-repellent”. Please see page 1 line 33-34 for the said changes.
Comment 4: L33 explain better “the temporal effect”, did you take samples at day 1 and then at day 7?
Reply: We thank the reviewer for highlighting this typographical error which is now rectified as “every day from day 1 to day 7”. Please see page 1, line 37-38 for the said changes.
Comment 5: L35 “…2% biochar application is the best combination…” of powder or granular biochar? For which soil?
Reply: We thank the reviewer for highlighting this important point. We have revised the text for more clarity as “2% powder biochar application rate is the best combination to improve the physicochemical properties of the tested mixed podzolic soil”. Please see page 1 line 38-39 for the said changes in the manuscript.
Comment 6: Keywords: do not include those words already included in the title, like physicochemical properties and podzolic soil
Reply: We agree with the reviewer comments and have deleted the redundant keywords from the list.
Introduction
Comment 7: L76 define “PAW”.
Reply: Abbreviation is defined at first use in the manuscript throughout the manuscript. text. Specifically, suggestions are followed as “Plant available water (PAW)” on page 2 line 67 in the revised manuscript.
Comment 8: L81-82: rephrase
Reply: Sentence is rephrased as “Determination of pore size and pore distribution pattern from different angle in biochar is essential to understand the effects on WHC and nutrient absorption capacity. Biochar characteristics like surface area and porosity depend on the temperature of pyrolysis and the raw materials used for biochar production [4, 39]”. Please see page 2 line 82-85 for the said changes.
Comment 9: L92 coarse?
Reply: Revised as “Sandy soil”. Please see page 2 line 96 for the said changes.
Materials and Methods
Comment 10: In general, M&M should be described much briefly, without the equations (with the reference is enough). Only topsoil and E-horizon where analysed, not the “mixed soil”. For the “treatments combinations” values of BD, TC, N have been calculated, not measured.
I do not think the experimental design, the methodology and the statistics are adequate and rigorous for a research paper.
Reply: We appreciate the perspective the reviewer poses for this comment. Before conducting the study, we reviewed more than 50 relevant articles from reputed journals. We followed standard procedure to execute experiments (provided references in all the sub-sections). We used three types of soil, two types of biochar with four rates and three replications for each experiment. We examined ten important physicochemical parameters to have large enough data points to improve the accuracy of the results and present a strong context for drawing meaningful interpretation for the data collected. For parameters like bulk density (BD), total carbon (TC), and nitrogen (N), we measured values in three types of soil and two types of biochar and then we followed references to calculate values of all treatment combinations of BD, TC and N. All other physicochemical parameters were carefully measured in the laboratory. In the study, we used high resolution SEM image for two types of biochar to know the pore size and biochar characteristics in terms of hydrophobicity or hydrophilicity. We used van Genucthen (VG) equation to know the value of permanent wilting point of the three types of soil and the treatment combinations. The significant difference of the treatment combination to control were provided in the table and text. We believe, we conducted a rigorous study based on tenets of the scientific methods. We executed experiments on ten important physicochemical parameters, three types of soil, two types of biochar with four rates and three replications for each using state of the art standard methodology published in the scientific literature. Furthermore, the results obtained were similar to many other reports in the scientific literature for other soil types (non-podzols), and these are mentioned in the discussion section. We believe the study findings are novel and would be helpful for the farmers seeking to apply biochar in their agricultural practices. In addition, podzol agriculture in boreal ecosystem is increasing due to expanded global population in these regions. We believe this work will be important in improving the knowledge in the scientific community of how biochar amendment can improve the physicochemical properties and quality of podzolic soils during agriculture production in boreal ecosystem.
Comment 11: L122 the soil you took for the “agricultural experimental station”, was subjected to any management practices?. Tillage? what crops where in the experimental station grown? Fertilization?
Reply: Soil samples were collected from agricultural experimental station sites and that land was used to cultivate experimental crops during the growing period in boreal climates. The management practices are consistent with those commonly applied in boreal climates like Corn and Maize for experimental purposes.
Comment 12: L125 what was the soil classification beyond “podzol”, which horizons did it have, at which depths… explain better “Topsoil and E-horizon soil samples were collected from 0-15 cm depth”
Clima classification? Rainfall? Mean temperatures?
Table 1 belongs to the result section
Reply: Please see the information in the methodology, where we briefly explained about the soil horizons in the page 3, line 134-139.
“During sampling we observed that in the sampling site, the average depth of topsoil was around 10 cm and E-horizon soil was 5 cm. Before cultivation, soil is usually plough and in time of plough, topsoil and E-horizon soil usually mixed in the agricultural field. Based on that, mixed soil prepared as a combination of topsoil and E-horizon soil with a ratio of 2 (topsoil): 1 (E-horizon soil) - according to the average soil layers in the field. Three soil types were evaluated: topsoil, E-horizon soil, and mixed soil”.
Soil samples were collected from the sampling sites and all the experiments were conducted in the laboratory, that’s why we did not mentioned any climatic status on the sampling site.
As indicated, we took the table 1 in the result section and revised the number of the sub-sections in the result section. Please see page 8 line 283 for the said change.
Comment 13: L137 soils where amended in the lab? Why not in the field? Explain better the experimental design
Fig 1. you do not have 720 experimental units
Reply: Thanks for the question. Soil samples were collected from the sampling site and all the experiments were conducted in the laboratory. Soil was sieved first in the lab to remove any stone and debris to have uniform mixtures for treatments and experiments were conducted carefully to get accurate results. The flow chart of experimental design described in the methodology section.
- As indicated, we have revised the experimental units in the flow chart (please see the page 4, line 155) and in the text as well.
Comment 14: L160 you did not analyse the “mixed soil”? and the values of sand, silt and clay from table 1?
Reply: We measured and calculated the particle size distribution of the three types of soil. The types of soils are topsoil, E-horizon soil, and mixed soil. We measured percentage of sand, silt and clay in the topsoil and E-horizon soil. In the table 1, sand percentage of mixed soil was 59 %. Topsoil sand 60% and E-horizon soil sand 58% and mixed soil sand percentage mean (60+60+58/3 = 59) because mixed soil was prepared with topsoil (2): E-horizon soil (1). Please see page 7 line before 273 for the percentage of sand, silt, and clay in the three types of soil.
Comment 15: L161 include the soil fractions sizes.
Reply: Soil fraction sizes were mentioned on that line. “calculated the particle size in the mixed soil according to the mixing ratio of topsoil (2): E-horizon soil (1). The percentages of clay (<0.002 mm), silt (0.002 – 0.06 mm), and sand (0.06 – 2 mm)”. Please see page 4 line 164 for the said changes.
Comment 16: L163 In what treatments was BD measured? Because in L167 you explain how you calculated BD for treatment combinations
Reply: We thank the reviewer for highlighting this important point. Bulk Density (BD) was measured for the three types of soil and two types of biochar. After that we used formula to calculate BD for all the treatment combinations in the rest of the experiments and we provided the references in the method section that we followed.
Comment 17: L164 I don´t think the way you measured BD is rigorous “tapped three times on the table”??? why 3 and not 5? In the reference you gave they do not do it like that [51] (Agriculture 2019, 9(6), 133; https://doi.org/10.3390/agriculture9060133): “Bulk density was measured by collecting undisturbed soil core samples and dividing the dry soil mass by the core volume“
Reply: We thank the reviewer for this insightful review. We read the suggested article again and revised the reference in the text “[52]” (please see page 5 line 174) (Soil System 2019b, 3, 49. doi:10.3390/soilsystems3030049). Authors in this published article explained that “After each addition of the mixture, the container was tapped down four times to achieve the desired BD”. In the experiment, we achieved desired BD with tapped down three time in the topsoil which was 1.31 gcm-3. This experimental BD density match with the reference BD on the sampling site.
Comment 18: L175 water repellency was then estimated in soil and biochar samples but not in the combinations?
Reply: We agreed with the reviewer’s comment. Water repellency (WR) was an estimated parameter. WR was estimated in all three types of soil and all treatment combinations following the standard procedure.
Comment 19: L183 I do not understand how conducting the WDPT at FC (I guess it is field capacity) and after 6 days drying at 28º in an oven (which will be close to permanent wilting point?) account to “measure temporal effects”
Reply: Water drop penetration time (WDPT) test was conducted following standard procedure. At first, we put a certain amount of sample in a cylindrical plastic container and saturated the sample for two days. After two days, we took the sample container from the water pot and facilitate drainage 48 hours. At that time, the plastic container covered to protect evaporation. Then we measured the field capacity of the samples and conducted the WDPT.
“After 6 days drying at 28ºC in the oven, the moisture content (maximum 3.35% and minimum 0.42% - described in figure 10) found below to the permanent wilting point (around 5% - 9%)”. We added the above text in the page 17, line 506-508.
Comment 20: L190 “from each biochar treat soils” means 3 soils x 2 biochar x 4 doses x 3 replications?
Reply: Yes, we conducted WDPT for the 72-treatment combination.
Comment 21: L199 if only sodium concentration was measured is not the CEC, and the other cations like K, Ca, Mg, Al, Fe?
Reply: Only sodium concentration was measured in our study. We used sodium acetate to know the value of CEC of the three types of soil, two types of biochar and all the treatment combination. The value of CEC was estimated for 72 treatment combinations.
Comment 22: L212 how did you calculate C and N? sometimes adding biochar or other amendments into the soil does not result in a proportional increase of TC and N, how did you account for priming effects? Mineralization rates might change
Reply: Following scientific procedure, the samples were ground well to make the particle size very fine and then Total carbon (TC) and nitrogen (N) were measured by elemental combustion analysis (CHNS Analyzer). We estimated percent of TC and N in three types of soil and two types of biochar. After that we calculated % of TC and N values for all the treatment combinations using 0, 0.5, 1 and 2% biochar rates. We described the details methodology of determination of TC and N in page 6 line 212-219.
Comment 23: L254 conducting one way anova you do not get the treatment interactions.
Reply: We have done Pearson T test to know the treatment effect and the significance of the data. We changed the word one way to general ANOVA in the manuscript. Please see page 7, line 262.
Comment 24: L258 explain better the control.
Reply: Control are the soil samples without having any types and rates of biochar. We added “only soil without biochar” in the text. Please see page 7 line 264 for the said change.
Results
Comment 25: Since most of the data for the "treatment combinations" have been calculated and not measured I do not think the results can be extrapolated to what would happen in real field conditions.
Reply: We thank the reviewer for this very insightful comment. The treatment combinations which were calculated and measured specifically mentioned in the methodology section. We calculated values of some physicochemical parameters of soil-biochar treatment combinations following measured values of soil and biochar of that specific parameter and after that used reference-based equation for calculating values of that parameter. The significance and comparison of the calculated values (of the treatment combinations) with other study findings were discussed in the discussion session (please see page 23 and line 672). This is a model that we hope this will be proved under field condition in future studies. In this study, all the experiments were conducted in the laboratory. For the further studies, one of the recommendations was made regarding experiments in the real field situation.
Comment 26: L313. You did not “observed” you calculated the theoretical BD for the “treatments combinations”
Reply: As suggested, we revised the word “found (from the calculated BD)”. Please see page 9 line 325 for the said change.
Comment 27: L315 of course you expected a high correlation between BD and biochar ratio since you have calculated BD proportionally to biochar rates
Reply: Thanks for this comment. At first, we measured bulk density (BD) of the three types of soil and two types of biochar. After that, we calculated BD of the soil-biochar treatment combination following reference-based equation. We found high correlation between BD and biochar ratio. We compared the values of BD with other study findings and that described in the discussion section with reference (please see page 22 and line 655-659).

Round 2
Reviewer 1 Report
Thank you for accepting my indications. I think the paper ready for publication.
Reviewer 3 Report
I am not satisfied with some of the answers the authors have given. For example:
Comment 22: L212 how did you calculate C and N? sometimes adding biochar or other amendments into the soil does not result in a proportional increase of TC and N, how did you account for priming effects? Mineralization rates might change.
Reply: Following scientific procedure, the samples were ground well to make the particle size very fine and then Total carbon (TC) and nitrogen (N) were measured by elemental combustion analysis (CHNS Analyzer). We estimated percent of TC and N in three types of soil and two types of biochar. After that we calculated % of TC and N values for all the treatment combinations using 0, 0.5, 1 and 2% biochar rates. We described the details methodology of determination of TC and N in page 6 line 212-219.
If TC and N are calculated, the priming effect is not taken into accout at all, which I consider is a major effect when adding organic residues into the soil